

# Seasonal drought prediction for semiarid northeast Brazil: About the added value of a process-based hydrological model

Tobias Pilz[1], José Miguel Delgado[1], Sebastian Voß[1], Klaus Vormoor[1], Till Francke[1], Alexandre Cunha Costa[2], Eduardo Martins[3], and Axel Bronstert[1]

[1]Institute of Earth and Environmental Science, University of Potsdam, Potsdam, Germany
[2]Institute of Engineering and Sustainable Development, University of International Integration of the Afro-Brazilian Lusophony (UNILAB), Acarape, Ceará, Brazil
[3]Research Institute for Meteorology and Water Resources – FUNCEME, Fortaleza, Ceará, Brazil

**Correspondence:** Tobias Pilz (tpilz@uni-potsdam.de)

**Abstract.** The semiarid northeast of Brazil is one of the most densely populated dryland regions in the world and recurrently affected by severe droughts. Thus, reliable seasonal forecasts of streamflow and reservoir storage are of high value for water managers. Such forecasts can be generated by applying either hydrological models representing underlying processes or statistical relationships exploiting correlations among meteorological and hydrological variables. This work evaluates and com-
pares the performances of seasonal reservoir storage forecasts derived by a process-based hydrological model and a statistical approach.

Driven by observations, both models achieve similar simulation accuracies. In a hindcast experiment, however, the accuracy of estimating regional reservoir storages was considerably lower using the process-based hydrological model, whereas the resolution and reliability of drought event predictions were similar by both approaches. Further investigations regarding
the deficiencies of the process-based model revealed a significant influence of antecedent wetness conditions and a higher sensitivity of model prediction performance towards rainfall forecast quality.

Based on the findings of this study, we recommend using a statistical approach for merely predicting reservoir level and drought events at regionally and monthly aggregated scales. However, for forecasts at finer scales of space and time or for the investigation of underlying processes, the costly initialisation and application of a process-based model is worthwhile. Further-
more, the application of innovative data products, such as remote sensing data, and operational model correction methods, like data assimilation, may allow for an enhanced exploitation of the advanced capabilities of process-based hydrological models.

# 1   Introduction

Drought is a type of natural hazard characterised by meteorological, hydrological, and water management conditions, affecting
many regions around the globe. Generally, it arises due to a shortage of water availability. A general valid or comprehensive



definition, however, is hardly achievable due to the many different possible causes, complex relationships and feedbacks among its determining factors and, consequently, different impacts on nature, society, and economy. As such, different categories can be distinguished ranging from meteorological (lack of rainfall) and hydrological (shortage of consumable water resources) to agricultural (water deficit for crops or husbandry) and socio-economic droughts (not enough of income to pay water price).

For the characterisation of droughts, different statistics can be computed describing duration, frequency, and severity based on various predictors and thresholds (Mishra and Singh, 2010).

The semiarid northeast of Brazil (NEB) is one of the world's most densely populated dryland regions (Marengo et al., 2017). Its climate is characterised by a short rainy season with high interannual variability. As a consequence, already since the colonisation in the 16th century, regularly occurring severe droughts causing famine and mass exodus have been reported.

Drought occurrence is primarily driven by Sea Surface Temperature (SST) anomalies in the eastern Pacific, i.e. the El Niño Southern Oscillation (ENSO), and the northern tropical Atlantic region (i.e. the Tropical Atlantic SST Dipole) influencing the location of the Inner Tropical Convergence Zone (ITCZ), which is the main source of rain during the rainy season in the NEB area (Hastenrath, 2012). Profound governmental actions for drought mitigation since the late 19th century resulted, among others, in the construction of thousands of small reservoirs and several large dams for water storage and provision within the

dry season and during dry spells. Still, severe drought events might endanger water supply, such as happening in the current series of drought years since 2012. Even the regular years (in terms of rainfall amount) of 2017 and 2018 were not able to eliminate or significantly alleviate water scarcity, still resulting in filling states of the largest reservoirs of less than 10 % (for the current state of water provision and statistics of the state of Ceará see http://www.hidro.ce.gov.br and the drought monitor http://msne.funceme.br). In addition, climate change is likely to aggravate water scarcity, calling for efficient strategies in the

management of water storages (de Araújo et al., 2004; Krol et al., 2006; Braga et al., 2013).

Reliable seasonal forecasting, i.e. forecasts of streamflow and reservoir storages for the upcoming rainy season, can be of significant value for water managers (Sankarasubramanian et al., 2009). Accurate precipitation forecasts over several months are still a challenge for dynamical climate models. However, many dryland regions are located in areas with distinct dry and rainy seasons, the latter often connected to large-scale atmospheric circulation patterns. Therefore, statistical models relating

meteorological or SST indices with streamflow or a combination of statistical and process-based models are applied in many dryland regions in the world to provide seasonal forecasts (e.g. Schepen and Wang, 2015; Seibert et al., 2017; Sittichok et al., 2018).

For the northern NEB region, the high correlation of rainfall and droughts with SST anomalies in the eastern Pacific and tropical Atlantic, together with correlation of pre-season rainfall offers a favourable setting for seasonal prediction (Souza Filho

and Lall, 2003; Sun et al., 2006; Hastenrath, 2012). Several studies exist for the area, typically employing one or several (realisations of) General Circulation Models (GCMs) driven by SST predictions, downscaled to a finer scale by statistical or dynamical downscaling approaches, whose meteorological (especially rainfall) outputs are eventually used as forcing in a hydrological model producing streamflow and/or reservoir level forecasts. For instance, Galvão et al. (2005), Block et al. (2009), and Alves et al. (2012) employed different hydrological models of varying complexity to generate streamflow and/or





reservoir level predictions. While model performance over daily timescales was generally reported to be low, over longer aggregation periods, such as at a monthly or seasonal scale, acceptable results could be achieved.

In a recent study, Delgado et al. (2017) investigated the use of a statistical relationship to provide seasonal reservoir level predictions. They used the two GCMs ECHAM4.6 and ECMWF, the meteorological output of each downscaled by three

different statistical approaches, generating ensembles of wet-season (i.e. January to June) hindcasts for each year in the period 1981 to 2014. Based on these meteorological hindcasts, they calculated a number of meteorological drought indices which are compared with observations to evaluate the skill of the predictions. Using reservoir storage as a target variable, they further computed hydrological drought indices and fitted a multivariate linear regression to predict these indices using the meteorological indices as predictors. Even though there was variation among the GCM and downscaling combinations, the

occurrence of meteorological drought could mostly be predicted with skill. Furthermore, their relatively simple statistical model was able to predict also hydrological droughts with skill. However, the absolute hindcast error was often not appreciably better than climatology, i.e. the observed long term average of a variable.

While being straightforward to apply and computationally advantageous, such statistical relationships, in contrast to process-based hydrological models, do not represent underlying processes and are less flexible in terms of the output variable and their

spatial and temporal resolution. However, what remains is the question, which approach produces more hands-on predictions for water managers and stakeholders in terms of accuracy, operability, and usability. As such, this study complements the work of Delgado et al. (2017), employing a process-based hydrological instead of a statistical model. Thus, the aim is to present and evaluate a forecasting system, predicting seasonal reservoir levels and the occurrence of hydrological droughts for the Jaguaribe river basin, located within the NEB region. Three different objectives are put to focus: First, the process-based hydrological

model and the statistical model of Delgado et al. (2017) shall be evaluated and compared in terms of reservoir level simulation performance. Second, the process-based hydrological model as an operational forecasting tool is to be verified in a hindcast experiment. Third, major sources of prediction and simulation errors in the modelling system are to be investigated. Thereby, the question whether the costly initialisation and use of a complex hydrological model is worthwhile in comparison to a much simpler statistical relationship is to be answered and guidelines for further research and the improvement of the forecasting

system shall be given.

## 2  Terminology

This study touches issues of Atmospherical Sciences, Hydrology and Water Resources Management. As terminology partially differs, the use of certain terms throughout the paper shall be clarified here. The word *forecast* generally refers to model based estimations of future meteorological or hydrological variables such as precipitation, streamflow, or reservoir level. The term

*prediction*, in this article like in many others, can be used synonymously to forecast. With *hindcast* we specifically denote retrospective forecasts, i.e. predictions issued for a period in the past building only on data available up to the time of start of the model run. The results are then compared with observations. In some occasions, the terms forecast and hindcast might be

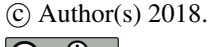



used interchangeably. In contrast to predictions or hindcasts, we denote *model simulations* as model runs driven by observations instead of forecasts of model forcing.

Many of the notions in the following sections will refer to the field of forecast verification. While being standard in Atmospherical Sciences, some terms are less common for the hydrological community and thus will be briefly explained in the

5 following. For more information, the reader is generally referred to textbooks such as Wilks (2005). The analysis of drought hindcasts will focus on their *quality*, i.e. the correspondence of such hindcasts with observations. This quality as defined by Murphy (1993) can be described in terms of nine different aspects of which five will be addressed explicitly in this study: *accuracy* as the average agreement of forecast–observation pairs which is as such inversely proportional to the *error*; *reliability* which, in the case of probabilistic drought forecasts, quantifies the average correspondence of forecast probabilities and

10 observed drought occurrences; *resolution* evaluating the ability of a model to correctly predict an event; *sharpness* describing the variability of forecasts of a model; and *skill* comparing the ability of a model with a much simpler reference model, such as climatology (which is the observed long term average of a specific variable) or persistence (i.e., no change of a variable or the pattern of a quantity over the forecast period).

Furthermore, we distinguish *process-based* from *statistical* models. The former are rather complex computer programmes

combining a set of mathematical equations (simple, linear up to complex differential equations), which can be derived from first order principles, e.g conservation of mass and energy. The aim here is to represent, up to a certain degree of abstraction, the governing sub-processes of the hydrological cycle and their interactions. They compute estimates of the unknown variables (e.g. river discharge, soil moisture, reservoir storage) as a reaction to a set of input or *driving* variables (e.g. precipitation, solar radiation, water abstraction). In this paper, the underlying process-based hydrological model refers to the WASA-SED model

which will be described later on. The latter, on the other hand, rely on purely empirical relationships between one or more predictors and the target variable, often consisting of only a single equation, typically obtained by regression. Consequently, the regression model of Delgado et al. (2017), which will be used for model intercomparison, will frequently be referred to as statistical model or statistical approach throughout this work.

## 3 Study site

The study area comprises the Jaguaribe river basin in the state of Ceará, northeast Brazil (see Fig. 1). The catchment is of crucial importance in terms of water supply for the whole state and has been intensively investigated in numerous studies (e.g. Bronstert et al., 2000; Gaiser et al., 2003; de Araújo et al., 2004; Krol et al., 2006; Mamede et al., 2012; van Oel et al., 2012; de Figueiredo et al., 2016). It covers an area of about 70,000 km$^2$ with a rural population of 2.7 million. Additionally, it is the source of water for the metropolitan area of Fortaleza with further 2.6 million people (IPECE, 2016). Annual precipitation sums

up to, on average, 755 mm per year whereas 90 % of rainfall occurs within the rainy season between January and June. Potential evapotranspiration is high with more than 2000 mm per year. The mean annual temperature is about 25 °C with little variation. Rainfall, however, is mostly convective with only a few events of high intensity per year and a strong inter-annual variation caused by SST anomalies resulting in a northward shift of the ITCZ inducing recurrent droughts that can last over several years





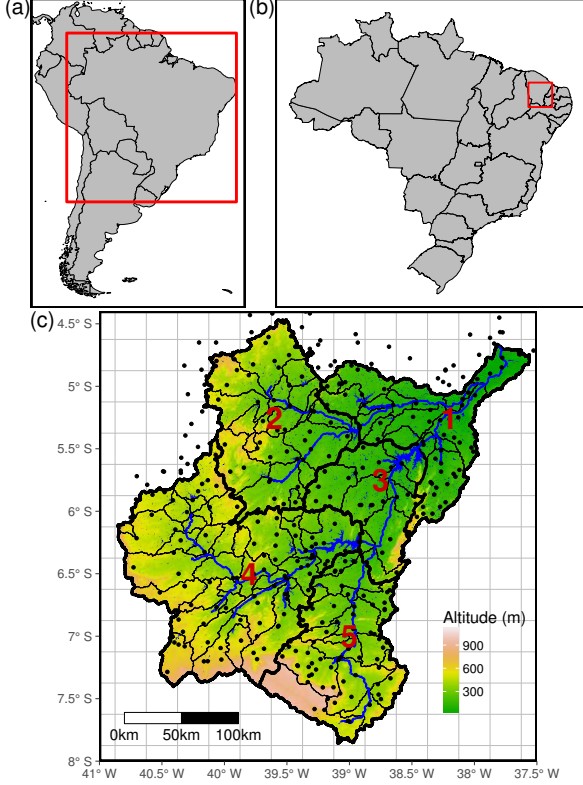

**Figure 1.** Overview over the Jaguaribe watershed (c) and location within Brazil (b) and South America (a). The five regions of interest (red numbers) are: 1) Lower Jaguaribe, 2) Banabuiú, 3) Castanhão, 4) Orós, and 5) Salgado. Thin black lines in c) outline subbasins, computational units within the model. Black dots are rainfall stations considered within the study. Background grid lines refer to the gridded meteorological dataset of Xavier et al. (2016).

(see also Sect. 1; Hastenrath, 2012; Marengo et al., 2017). As the geology is characterised by a primarily crystalline basement with low-density fractures, water supply needs to be secured by surface water resources. Accordingly, thousands of small and several large reservoirs were constructed. The small reservoirs are typically bordered by uncontrolled earth dams, mainly serving for water provision of rural population and livestock. Conversely, large so-called *strategic reservoirs* contain a barrage with intake devices for active regulation, are sometimes also used for hydropower production, and serve as water ressources for larger towns and cities and industrial farming. These settings cause meteorological droughts (lack of precipitation) and hydrological droughts (lack of surface water) to be often out of phase (de Araújo and Bronstert, 2016; van Oel et al., 2018).

For the present study, the Jaguaribe catchment was subdivided into five sub-regions, named after the main tributary river or the major reservoir at its outlet: Banabuiú, Orós, Salgado, Castanhão, and Lower Jaguaribe (see Fig. 1 for their location).



## 4 Data and Methods

### 4.1 General workflow

The aim of this study is to elucidate the application potential of a process-based hydrological model for water ressources and drought prediction. Consequently, hindcasts of reservoir volumes and hydrological drought indices shall be produced, driving
the model by meteorological hindcasts. The general workflow is illustrated in Fig. 2.

    A process-based hydrological model was first calibrated to observations and an initial model run conducted for the period of 1980 until 30 June 2014. This initialisation run was driven by observed meteorology and at each 1 January the storage volume of each strategic reservoir was replaced by the observed value. Furthermore, if available, measured reservoir releases through a dam's intake devices were fed into the model in order to make use of as much information as available to produce
simulations as realistic as possible. The first year of the run was used as a warm-up to bring the model states into equilibrium. At each end of year, the model's state variables, including soil moisture, groundwater, river, and small (i.e. non–strategic) reservoir storages, were stored. This entire procedure is intended to mimic the conditions in a real forecast situation. In a specific hindcast run, the model was then re-initialised with the saved model states and driven by hindcast meteorology. These runs were conducted for the wet seasons (1 January to 30 June) of 1981 to 2014. The resulting strategic reservoir volumes
were used to infer drought indices which were evaluated employing verification metrics. To distinguish uncertainties from the meteorological hindcasts and in order to investigate mere model performance, the model runs were performed in two ways: driven by observations (*simulation runs*) and meteorological hindcasts (*hindcast runs*). In order to identify the strengths and weaknesses of the hydrological model, the results of the simulation runs were further analyzed. In this context, the model output (reservoir storage) was stratified. The details of the individual processing steps are described in the following.

### 4.2 Data

To parametrise the hydrological model, various spatial data were obtained including a 90 m x 90 m SRTM digital elevation model (DEM), a soil map provided by the research institute for meteorology and water resources of the state of Ceará (FUNCEME) along with soil parameters from a local database (Jacomine et al., 1973) from which the necessary model parameters were calculated employing pedo-transfer functions, a landcover map from the Brazilian Ministry for the environment with
parametrisations assembled by Güntner (2002), and a map of small and strategic reservoirs provided by FUNCEME. Reservoir parameters were made available by the Company for Water Resources Management of Ceará (COGERH) and FUNCEME and include the year of dam construction, storage capacity, and water level–lake area–storage volume relationships along with daily resolution time series of water levels and artificial water release. A time series of daily precipitation for 380 stations within and in close vicinity around the study area were provided by FUNCEME. Other daily meteorological time series needed by
the model (relative humidity, air temperature, and incoming shortwave radiation) were derived from the gridded dataset (0.25° × 0.25° resolution) of Xavier et al. (2016).





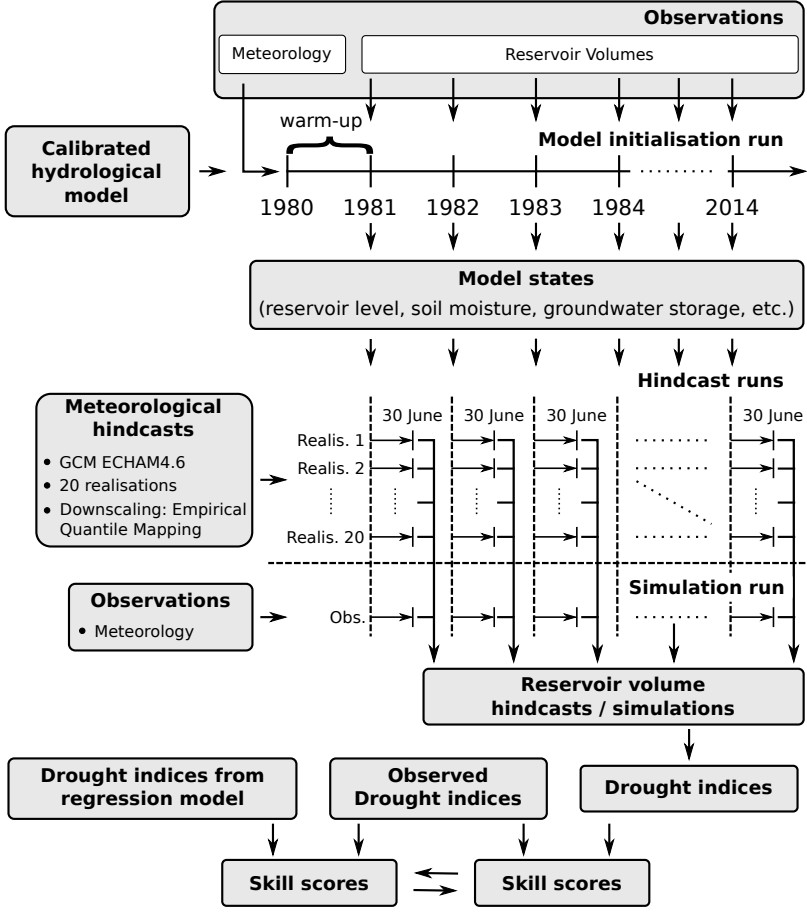

**Figure 2.** Workflow for the generation and evaluation of hindcasts of hydrological drought indices.

## 4.3 Meteorological hindcasts

Daily meteorological hindcast data for the period 1981 to 2014 used as input into the hydrological model stem from an ensemble (20 members) of ECHAM4.6 GCM runs (Roeckner et al., 1996) which were bias corrected by Empirical Quantile Mapping (EQM) (Boé et al., 2007; Gudmundsson et al., 2012). Although Delgado et al. (2017) identified some deficiencies regarding
5  this product, there was no clear better performing alternative. In addition, ECHAM4.6 is already employed operationally by the local water authority FUNCEME (Sun et al., 2006), and, in contrast to other seasonal forecast systems like those by ECMWF, it comes without further costs for operational use, making it the candidate for future operational application. The 20 member ensemble runs of ECHAM4.6 were conducted and results provided by FUNCEME. More information are given in Delgado et al. (2017).





### 4.4 Hydrological modelling

#### 4.4.1 The WASA-SED model

The hydrological model WASA-SED, revision 257, was employed for the hindcasts of reservoir volumes. WASA-SED is a deterministic, process-based, semi-distributed, time-continuous hydrological model. The representation of hydrological processes

focuses on dryland environments. A complex but efficient hierarchical spatial disaggregation scheme allows for application over large scales up to an order of magnitude of 100,000 $\mathrm{km}^2$ (Güntner and Bronstert, 2004; Mueller et al., 2010). Reservoirs can be simulated by treating large strategic reservoirs in an explicit manner while representing smaller ones as lumped water bodies of different size classes to efficiently account for water retention of many small reservoirs in a study region (Güntner et al., 2004). The model was developed for and successfully applied in the semiarid areas of northeastern Brazil (Medeiros

et al., 2010; Krol et al., 2011; de Araújo and Medeiros, 2013; Medeiros et al., 2014) and used for other dryland regions, such as in India (Jackisch et al., 2014) and Spain (Mueller et al., 2009, 2010; Bronstert et al., 2014).

#### 4.4.2 Model parametrisation and calibration

The model was parametrised using the lumpR package for the statistical environment R (Pilz et al., 2017). This included the delineation of catchment and model units, assembly, calculation, and checking of parameters, and the generation of the

15 model's input files. Meteorological data were interpolated to the respective spatial units (sub-basins). For rainfall, this step used the Thiessen Polygon method as implemented in the Information System for Water Management and Allocation (SIGA) (Barros et al., 2013). For the other meteorological variables, Inverse Distance Weighting (IDW) from the R package geostat (Kneis, 2012) was used. Reservoir data were processed and prepared for the model. A total of 36 strategic reservoirs within the study area was selected for explicit treatment in the model according to their size and importance for water management.

The model was calibrated independently for each of the five regions in the study area (see Fig. 1). Calibrated output of upstream regions was used as boundary condition for downstream regions. Due to lack of data for Lower Jaguaribe, the calibrated parameters of Castanhão were transferred. However, sufficient data were available for further analyses. The calibration period spanned 2003 to 2010, which includes both wet (2004 and 2009) and dry (2005, 2007, and 2010) years.

Daily reservoir volume variation of the strategic outlet reservoir of a specific region was used as target variable as reservoir

level measurements were assumed to be more reliable than streamflow observations. Streamflow in the area is highly variable and rivers, especially in the downstream part of the catchment, are characterised by broad and dynamic cross sections and dense riparian vegetation inducing large uncertainties in streamflow measurements derived from rating curves. To minimize uncertainties due to strategic reservoir management, only positive variations (i.e. net reservoir volume gain) were considered whereas a net loss was set to zero during calibration. However, for region Salgado, streamflow measurements had to be used

as this specific region does not contain a strategic reservoir at its outlet.





In total, 15 parameters were chosen for calibration. As objective function, a modified version of the Nash–Sutcliffe Efficiency (NSE) called Benchmark Efficiency (BE) following Schaefli and Gupta (2007) was employed. It is calculated as

$$BE = 1 - \frac{\sum_{t=1}^{N}(q_{obs}(t) - q_{sim}(t))^2}{\sum_{t=1}^{N}(q_{obs}(t) - q_{bench}(t))^2} \tag{1}$$

with $t$ being the index of time containing $N$ time steps within the calibration period, $q_{obs}$ are the observations, $q_{sim}$ the

simulations, and $q_{bench}$, instead of being the average of the observations as in the traditional NSE, represents the mean of the observations for every Julian day over all years within $N$ (i.e. the mean annual cycle). In this way, a value of $BE > 0$ means the model is able to reproduce the average yearly dynamics better than simply using statistics. Consequently, a value of $BE = 1$ signifies perfect agreement of simulations with measurements. Eventually, BE as performance measure employs a much stricter criterion on simulated hydrological dynamics compared to using the NSE measure.

For calibration, the Dynamically Dimensioned Search (DDS) algorithm (Tolson and Shoemaker, 2007) implemented in the R package ppso (Francke, 2017) was used. Since DDS was developed for computationally demanding hydrological models it is able to obtain satisfying results within the order of 1,000 to 10,000 model calls. For this study, the number of calls was limited to 5,000 for every region which resulted in about 10,000 hours of CPU core processing time on a high performance cluster.

### 4.4.3 Analysis of simulation performance and influencing factors

To investigate different aspects of model performance, a set of goodness measures common in the context of hydrological modelling was chosen. This set consisted of the Kling–Gupta Efficiency (KGE) and its three components correlation, bias, and deviation of standard deviations of simulations and observations, see upper part of Table 1. Like NSE and BE presented above, KGE scales from minus infinity to one where one is the optimum value achieved for maximum correlation (i.e. $COR = 1$) and no deviation of means and standard deviations. To assess which factors influence the model performance, several candidate

descriptors where selected which are presented in the lower section of Table 1. These descriptors were tested for their capability to explain model performance in time and space in a regression approach by using these descriptors as predictors and the performance metrics as the response variable.

For the analysis, the calibration period 2003 to 2010 was used. Each response variable (i.e. performance metric) was calculated for each of the 36 strategic reservoirs located in the study area. Furthermore, each year was divided into a falling period, where the difference of reservoir levels for two consecutive days was negative, and a rising period where the difference was

greater than or equal to zero. For each reservoir, year, and period, the respective performance was computed and analysed separately. This resulted in a total of 32 reservoirs *times* 8 years *times* two periods *minus* some missing observations, i.e. 484 values to be aggregated for each response variable. The predictors were either static and unique for each reservoir ($A_{up}$, $V_{cap}$, $n_{resup}$), region specific and dynamic as aggregation over a certain amount of time ($P_{max}$, $P_{reg}$, $P_{12}$, $P_{36}$) or a grouping variable by

itself ($\Delta_{vol}$) (see Table 1 for the meaning of symbols).

To identify predictor importances and their influence on the performance measures, a regression tree analysis was conducted using the R package rpart (Therneau and Atkinson, 2018) based on Breiman (1984). This technique was chosen to allow for skewed data distributions, non-monotonous and threshold behaviour and interaction between the predictors, while still provid-



**Table 1.** Response and predictor variables used for the analyses of the process-based model performance.

| Abbrev. | Explanation |
| --- | --- |
| **Responses** | |
| KGE | Kling–Gupta Efficiency (Gupta et al., 2009): $1 - \sqrt{(\text{COR} - 1)^2 + \text{BIAS}^2 + \text{VAR}^2}$ |
| COR | Pearson correlation of simulations $y$ and observations $o$: $\frac{cov(y,o)}{\sigma_y \sigma_o}$ with $cov$ being their covariance and $\sigma$ their standard deviations |
| BIAS | Deviation of means $\mu$: $\frac{\mu_y}{\mu_o} - 1 \in [-1, \infty)$ |
| VAR | Deviation of variability: $\frac{\sigma_y}{\sigma_o} - 1 \in [-1, \infty)$ |
| **Predictors** | |
| $A_{up}$ | Upstream catchment area of the reservoir ($km^2$) |
| $V_{cap}$ | Reservoir volume capacity ($hm^3$) |
| $n_{resup}$ | Number of upstream reservoirs (-) |
| $\Delta_{vol}$ | Rising or falling period of reservoir volume (-) |
| $P_{max}$ | Maximum regional daily precipitation sum over rising / falling period of a year ($mm$) |
| $P_{reg}$ | Regional precipitation sum over rising / falling period of a year ($mm$) |
| $P_{12}$ | Regional precipitation sum over the entire previous year ($mm$) |
| $P_{36}$ | Regional precipitation sum over 36 months of the preceding years ($mm$) |

ing interpretability of the results, i.e. tree structure. For each response variable, an individual tree was built (for illustration, see resulting tress in Sect. A). Except for $\Delta_{vol}$ (categorical), each predictor and response variable was treated as numerical. The partitioning within the regression tree was carried out using the ANalysis Of VAriance (ANOVA) method. Tree size was influenced by setting the splitting threshold (i.e. the minimum increase in $R^2$ to accept a partitioning) to a value of 0.01 and

the minimum number of values in a terminal (or *leaf*) node to 40, which was equal to about 8 % of the initial dataset. The most important predictors for a certain response were then distinguished by calculating relative importance measures with the rpart package. The results for all predictors were then scaled to sum up to 100. In order to get an impression of the concrete effect of each predictor, the two leaf nodes with the highest and lowest median response values for each tree, respectively, were identified (e.g. nodes 8 and 3 in Fig. A1). For these two nodes, the ranges of each numerical predictor (except $\Delta_{vol}$) were

classified into four groups ranging from small to large to facilitate visual investigation.

## 4.5   Drought hindcasting

### 4.5.1   Hydrological drought quantification

As water stored in surface reservoirs is of primary importance to water supply, hydrological drought indices based on surface reservoir filling level appear to be the most adequate choices to identify and characterise hydrological droughts in the study

area. Thus, for the quantification of hydrological droughts, the regionally and monthly aggregated reservoir storage was chosen





as drought indicator:

$$I_t = \frac{\sum_{i=1}^{R_j} V_t^i}{\sum_{i=1}^{R_j} V_{cap}^i} \tag{2}$$

with $t$ being the time index, $V_t^i$ the volume stored in reservoir $i$ of a certain region $R_j$ (i.e. one of the five sub-regions of interest illustrated in Fig. 1), and $V_{cap}^i$ the storage capacity of that reservoir. This metric was calculated for each of the five regions of interest ($R$) and each month of the hindcast period (wet seasons, January to June, of 1981 to 2014). For each month the last daily value was taken.

A drought was then defined as

$$D_t = \begin{cases} 1 & \text{if } I_t < q_{0.3} \\ 0 & \text{if } I_t \geq q_{0.3} \end{cases} \tag{3}$$

where $D = 1$ denotes drought, $D = 0$ indicates no drought, and $q_{0.3}$ is the 0.3 quantile of $I$ over the hindcast period. The definition of $q_{0.3}$ is based on the choice of local decision makers who defined this value as warning threshold for reservoir scarcity. The threshold was applied to each region individually and, thus, resulted in regionally different drought thresholds. As such, the results of this study will be comparable to the work of Delgado et al. (2017).

### 4.5.2 Verification of drought hindcasts

Hindcasts of reservoir volumes ($V_t^i$) and, consequently, the hydrological drought index ($I_t$) were verified employing the Root Mean Square Error (RMSE), the Relative Operating Characteristic Skill Score (ROCSS), and the Brier Skill Score (BSS). Definitions and discussions of the various forecast verification metrics can be found in textbooks such as Wilks (2005). In the following, short explanations for each selected measure shall be given.

The RMSE is a deterministic measure and was derived by calculating the root of squared differences of hindcasts and observations averaged over all values of the hindcast period:

$$RMSE = \sqrt{\frac{1}{N} \sum_{t=1}^{N} (I_t^f - I_t^o)^2} \tag{4}$$

where $N$ is the number of forecasted time steps and the superscripts $f$ and $o$ denote forecast and observation, respectively. It was calculated multiple times by using as $I_t^f$ each GCM member individually and, in addition, the median of members as deterministic value. The metric quantifies the average magnitude of hindcast errors in units of the target variable, i.e. in this case regional reservoir storage in percent points, and is therefore useful for the interpretation of suitability of the model for water managers who rely on accurate forecasts of volumes to coordinate reservoir operation. As such, the RMSE refers to the attribute of accuracy. The lower the RMSE, the lower the forecast error and the higher the accuracy.

The Relative Operating Characteristic Skill Score (ROCSS) quantifies the ability of a model to correctly discriminate between events and non-events. In this context, an event is defined as a hydrological drought which, in turn, is distinguished by





the drought index falling below the 0.3 quantile ($q_{0.3}$) as explained above. The ROCSS is based on the ROC curve which plots the probability of event detection against the false alarm rate for different thresholds of forecast probability defining an event. Taking the Area Under the Curve (AUC) of this graph, the skill score can be calculated as

$$ROCSS = 2 \cdot AUC - 1 \tag{5}$$

The value ranges between $-1$ and $1$ with values lower than or equal to zero indicating the false alarm rate being greater than or equal to the probability of event detection and, thus, the model having no skill. A value of one represents the highest score, i.e. the model is able to predict every event and non-event correctly. As such, the ROCSS is a measure for event resolution of probabilistic forecasts.

The Brier Score (BS) measures the mean squared error of probabilistic forecasts and indirectly contains information about
reliability, resolution, and the variability of observations (the latter being commonly referred to as uncertainty). As such it can be calculated as

$$BS = \frac{1}{N} \sum_{t=1}^{N} (D_t^f - D_t^o)^2. \tag{6}$$

The corresponding skill score (BSS) compares the BS of a forecast model with that of a simple reference forecast, in our case climatology:

$$BSS = 1 - \frac{BS}{BS_{reference}} \tag{7}$$

with $BS_{reference} = 0.3$, corresponding to $q_{0.3}$, the initially defined long-term average probability of drought occurrence (as described above). It follows that $BSS \in (-\infty, 1]$ and a forecast model having skill relative to the reference model if $BSS > 0$.

### 4.5.3  Comparison with results from a statistical model

A goal of this study is to answer the question whether a complex process-based simulation model is worthwhile in comparison
to a much more convenient statistical approach. To achieve this, results from this study shall be compared with the second approach (M2) of Delgado et al. (2017). We compare the process-based model to the multivariate linear regression (MLR), which generated hindcasts of the same hydrological drought index in the same area of investigation using the same data. As predictors, meteorological drought indices were employed. To compare the mere simulation performances, both models were first driven by observed meteorology to exclude the effect of the downscaled GCM runs. In a second step, the two approaches
were evaluated for real operational hindcasts.

## 5  Results

### 5.1  Comparison of model performance in simulation mode

Figure 3 compares the performances of the process-based and statistical model in simulating relative regional reservoir storage driven by observed meteorology. The regional RMSE varies between 5 and 18 percent points whereas for the whole catchment



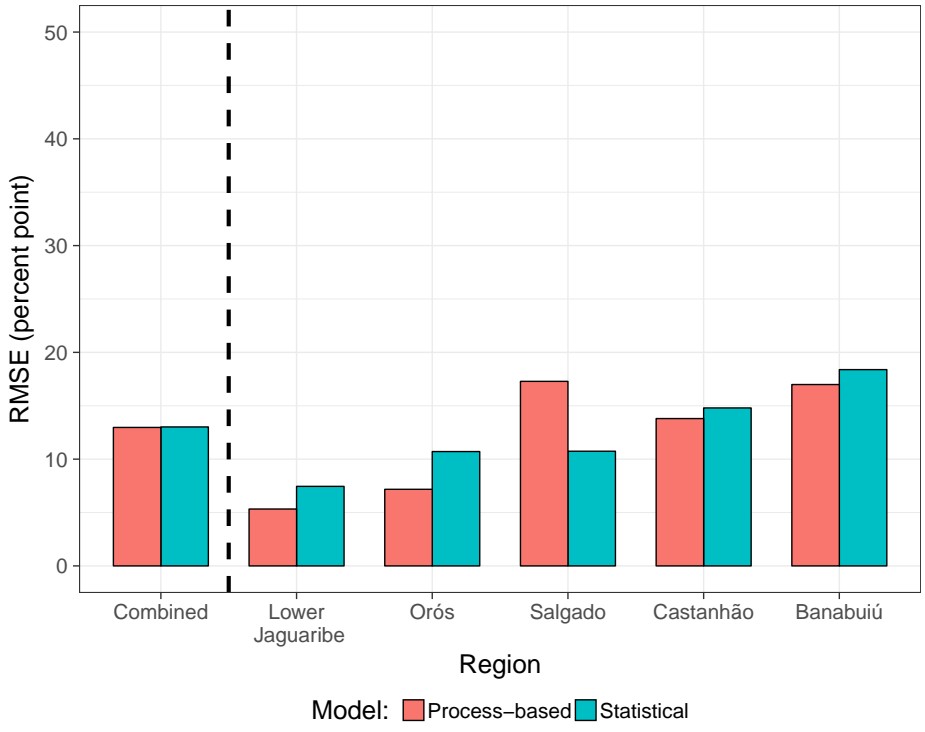

**Figure 3.** Comparison of accuracy in predicting relative reservoir storage for the process-based and statistical model run with observed forcing.

both modelling approaches achieve a result of about 13 percent points. Overall, the performance differences between the two models are small for all regions. Only for Salgado the statistical model shows a lower RMSE compared to the process-based model and the difference among the two approaches is largest (6 percent points). For all other regions, the process-based model exhibits a slightly higher accuracy, and the inter-regional ranking is equal for both approaches. Generally speaking, both

5    models show a comparable performance, suggesting they are equally suitable for their application in hindcast mode.

## 5.2    Comparison of model performance in hindcast mode

The uppermost panel of Fig. 4 shows that in hindcast mode, the accuracy in terms of RMSE considerably decreases when compared to simulation mode for both types of models. However, in contrast to the situation in simulation mode, the statistical approach outperforms the process-based model for all regions. While for the statistical model, deterioration in terms of RMSE

10    is generally less than 10 percent points, the process-based model achieves significantly lower accuracy with increasing RMSE by up to almost 30 percent points. This degradation of model performance in hindcast mode for the process-based model is especially pronounced for the Banabuiú region.



The lower two panels of Fig. 4, however, demonstrate that both approaches are able to generate drought hindcasts with skill. The resolution of event hindcasting of the two models (i.e. the ROCSS) is very similar combined over the whole catchment. Regional differences are more pronounced but still negligible. For some regions the process-based, for other regions the statistical model performs slightly better. The BSS, while also indicating skill, shows lower performance values which can be attributed to lacks in accuracy (as already indicated by RMSE) and reliability.

An attribute plot, as the one presented in Fig. 5, can reveal more details on that issue (Wilks, 2005). Therein, the predicted probability of drought occurrence (obtained from the outcomes of individual ensemble members) is plotted against the relative frequency of observed drought occurrences (solid lines) together with the relative prediction frequency of a certain forecast probability interval (dotted lines). It demonstrates several verification attributes including resolution (the flatter the solid lines, the less resolution), reliability (agreement with the gray diagonal line), sharpness (dotted coloured lines), and skill (values within the gray region contribute positively to BSS). Apparently, predictions from both models contain skill except for low forecast probabilities where both models contribute negatively to BSS. Furthermore it can be seen that both approaches exhibit problems in terms of reliability. Specifically, forecast probabilities are too low compared to observed occurrences, which is generally denoted as underforecasting. This observation appears to be a bit more pronounced for the process-based than for the statistical model. That also holds true for sharpness, as the statistical approach shows slightly more confidence for higher forecast probabilities, i.e. the relative frequency of maximum forecast probability is higher.

## 5.3 Model performance attribution

### 5.3.1 Hindcasts

The monthly aggregated accuracy of the hindcasts, i.e. performance with increasing lead time, is shown in Fig. 6. Overall, the hindcast error (i.e. RMSE) increases with lead time (i.e. progression of the wet season), even when using observed forcing. The statistical approach generally produces better hindcasts. Its RMSEs differ only little from runs with observed forcing. Also the increase of RMSE with lead time is very similar. For the process-based model, hindcasts deviate clearly from observation based results (as was already shown in Fig. 4). The error increases much stronger over the hindcast horizon. However, its RMSE values reach a plateau at about 40 percent points in March. Generally, it can be seen that aggregating the ensemble members by using the median of reservoir storage hindcasts (solid lines) is usually a better choice than most of the single ensemble members (distributions shown as boxplots). The spread of ensemble member results differ for the two approaches. These ranges are clearly larger for the process-based model in January and February, but comparable for the other months.

In Fig. 7 prediction accuracy is assessed for different wetness conditions (i.e. dry, normal, wet) over different accumulation time periods for rainfall. Again, when driven by meteorological hindcasts, the statistical approach performs best with relatively small differences to results obtained using observed forcing. Under wet conditions, irrespective of the rainfall accumulation period, the error is highest for most settings. The only exception from this pattern shows the process-based model driven by hindcasts. Here, the error under dry preconditions increases with increasing rainfall accumulation length while the performance under wet preconditions improves with longer accumulation length.





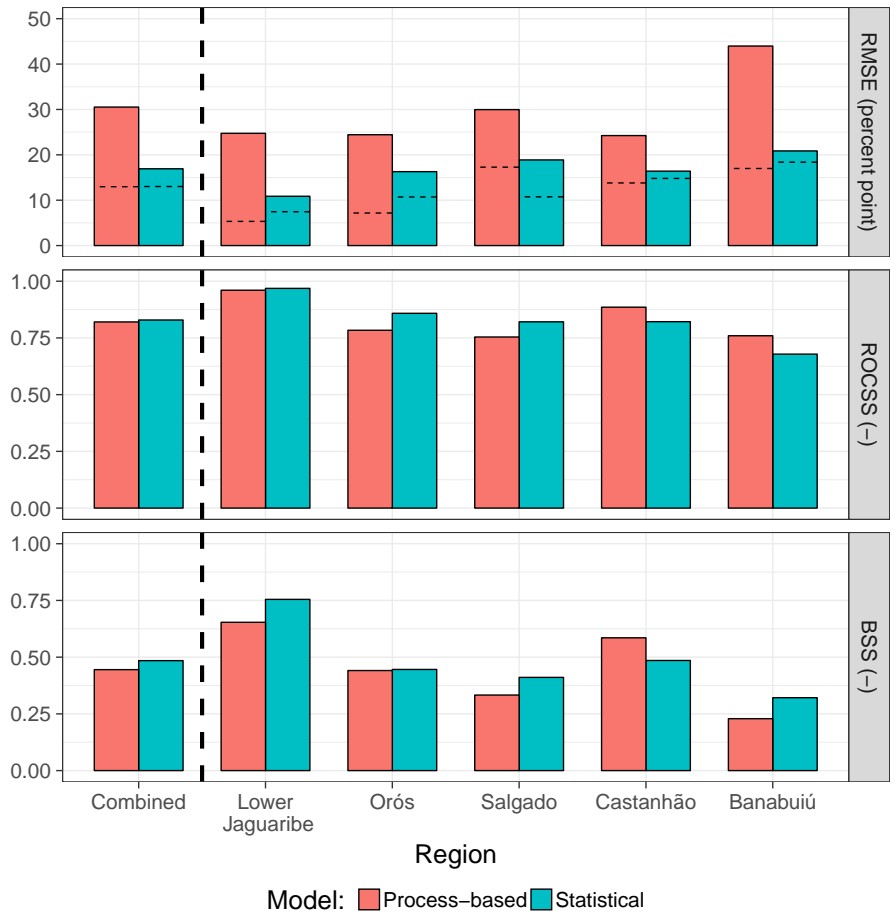

**Figure 4.** Model performance in hindcast mode for the two model approaches. In the top panel, horizontal dashed lines in the bars mark the results obtained with observed forcing ("simulation mode", as in Fig. 3). Note that for RMSE low values indicate a better performance while for ROCSS and BSS higher values are favoured.

### 5.3.2 Process-based simulation performance

In the preceding subsections it was shown that the process-based model does not outperform the statistical approach. Moreover, in hindcast mode, the process-based model often achieved worse performance measures, especially in terms of accuracy. This subsection therefore aims at the identification of deficit causes by analysing the results of process-based model calibration and potential influencing factors of simulation performance in more detail.

Regional calibration performance of the process-based model is summarised in Table 2. A good overall agreement of simulated and observed reservoir dynamics in terms of BE values could be achieved during calibration. However, PBIAS as a performance metric not used in the calibration shows, on the one hand, acceptable values of no more than 12 % but, on the



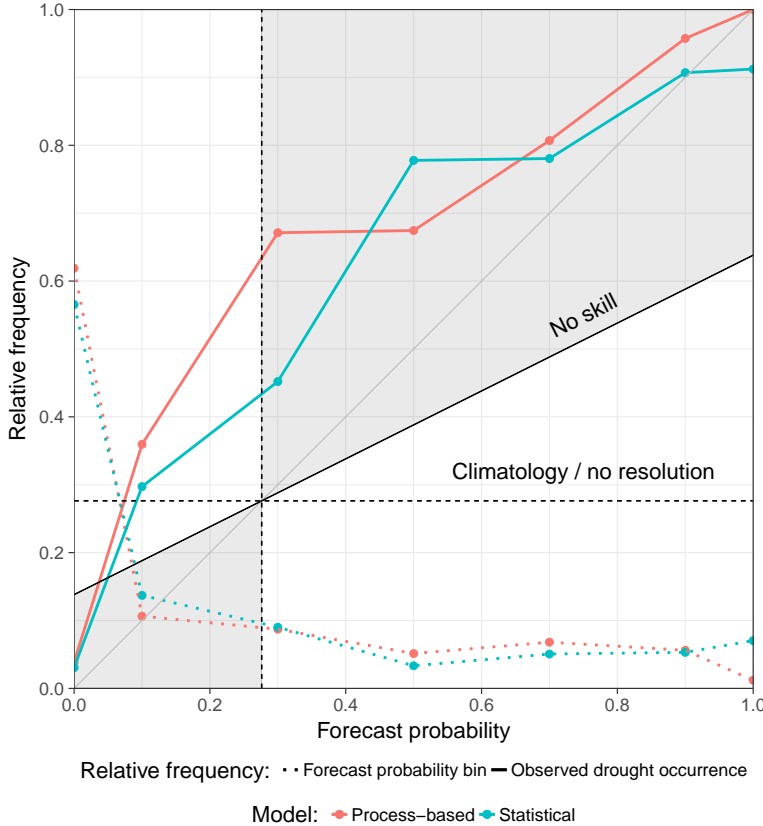

**Figure 5.** Attribute plot of drought hindcasts aggregated over the whole study area. Values within the gray region contribute positively to the Brier Skill Score (BSS). For further information on the reading of attribute plots see text or consider textbooks on forecast verification, such as Wilks (2005).

other hand, a consistent slight overestimation of reservoir level dynamics. It can be further observed that a good BE value does not correlate with a low PBIAS.

The individual regression trees of the analysis of process-based model performance for each response variable are shown in Appendix A. Herein, the findings from the trees shall be summarised. Table 3 shows the relative importance of each of the potential predictors in explaining the variability of model performance. Apparently, overall model performance (KGE) primarily depends on the wetness preconditions. While reservoir size ($V_{cap}$) and catchment size ($A_{up}$) play only a minor role for the overall performance metric KGE, they clearly affect correlation and bias. (Mis-)match of standard deviation (VAR), however, is mainly determined by both wetness conditions and reservoir size. The current rainfall conditions in terms of intensity ($P_{max}$) and sum over a rising / falling period ($P_{reg}$) as well as whether it is a reservoir level increase or decrease period ($\Delta_{vol}$), and the number of upstream reservoirs ($n_{resup}$), show little or no explanatory value for any of the performance measures.





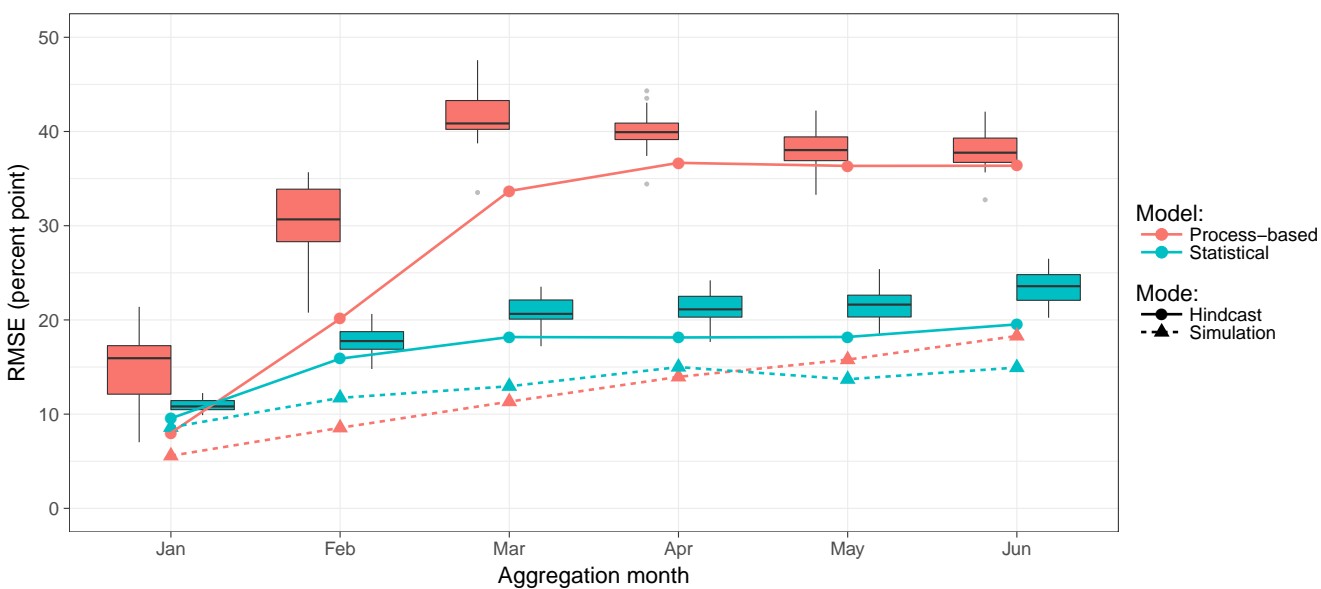

**Figure 6.** RMSE of regional reservoir storage hindcasts with increasing forecast horizon / lead time. Each box reflects the distribution of the 20 ensemble members. Coloured solid lines refer to the ensemble median taken as deterministic forecast and analysed individually.

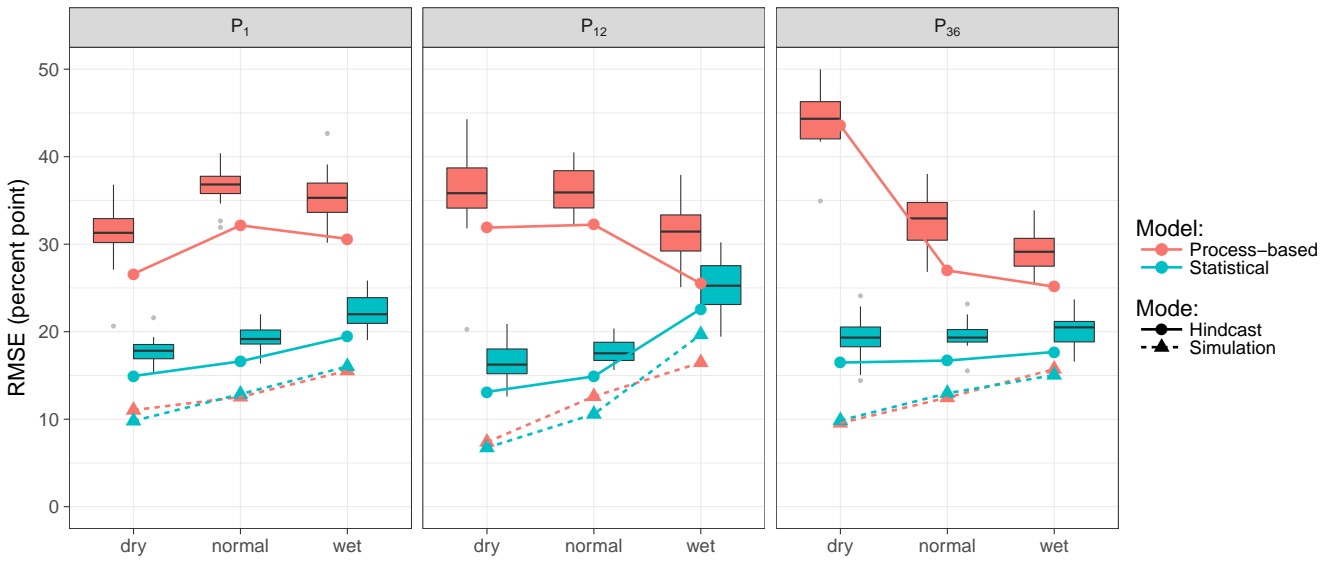

**Figure 7.** RMSE of regional reservoir level hindcasts for different antecedent wetness conditions. Wetness is expressed by three different accumulation horizons of rainfall (1, 12, and 36 months; left, centre, right). Each box reflects the distribution of the 20 ensemble members. Coloured solid lines refer to the ensemble median taken as deterministic forecast and analysed individually.





**Table 2.** Results of regional calibration of the process-based model. BE refers to *Benchmark Efficiency* (Eq. 1) and was used for calibration; PBIAS is *percent bias*, i.e. the average tendency for over- or underestimation of simulations in comparison to observations. For Lower Jaguaribe, no observations at the catchment outlet were available.

| Region | BE | PBIAS (%) |
|---|---|---|
| Banabuiú | 0.84 | 11.78 |
| Orós | 0.76 | 0.92 |
| Salgado | 0.79 | 7.37 |
| Castanhão | 0.76 | 6.93 |
| Lower Jaguaribe | [no obs.] | [no obs.] |

**Table 3.** Relative predictor importance for each response variable. Values are rounded to integer values. Font sizes of numbers reflect the importance value.

| Response | Relative predictor importance (%) | | | | | | | |
|---|---|---|---|---|---|---|---|---|
| | $V_{cap}$ | $A_{up}$ | $n_{resup}$ | $\Delta_{vol}$ | $P_{max}$ | $P_{reg}$ | $P_{12}$ | $P_{36}$ |
| KGE | 0 | 14 | 0 | 0 | 1 | 1 | 28 | 55 |
| COR | 60 | 14 | 0 | 0 | 1 | 2 | 2 | 20 |
| BIAS | 52 | 22 | 15 | 0 | 2 | 2 | 3 | 5 |
| VAR | 24 | 15 | 7 | 0 | 5 | 4 | 13 | 31 |

To analyse the specific influence of predictors on the response variables, Fig. 8 relates the values of the most influential predictors to the corresponding performance measures. This is done by plotting the occurrences of predictor categories in the highest and smallest valued leaf nodes of the regression trees. It shows that under dry preconditions ($P_{36} = \min$) there is a tendency for underestimation of standard deviations ($VAR = \min$), i.e. a less variable reservoir storage series than observed, but a better correlation between simulations and measurements ($COR = \max$) and a better overall performance ($KGE = \max$). On the other hand, under wet conditions, especially for larger reservoirs ($V_{cap} = \max$), results tend to show an overestimation of variability ($VAR = \max$). For small reservoirs correlation is mostly low. It should be noted, however, that relationships cannot always be clearly distinguished. For instance, a low precipitation sum over the preceding year ($P_{12}$) may result in both a high and a low KGE value whereas very low precipitation over the preceding 3 years ($P_{36}$) only led to a high KGE.





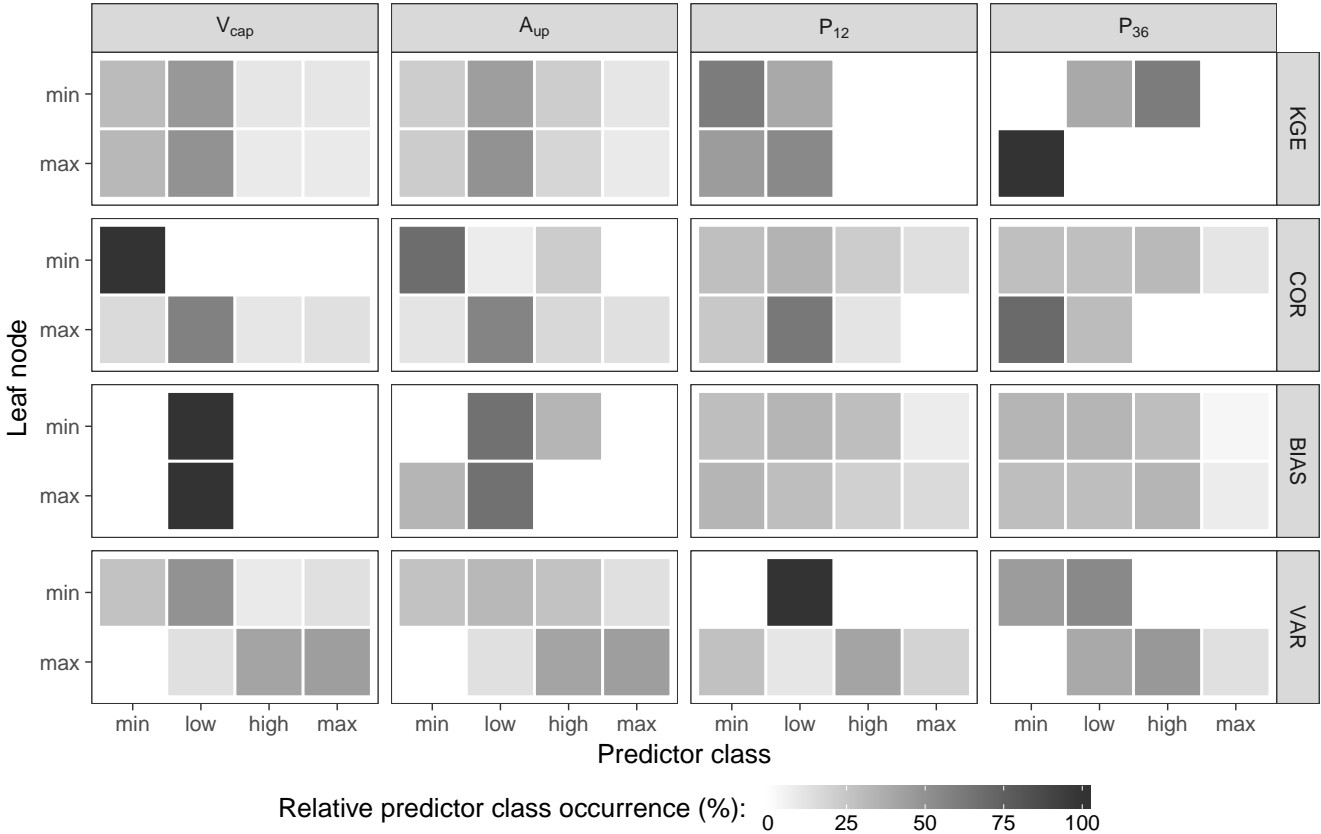

**Figure 8.** Relative distribution of predictor class occurrences within the largest and smallest valued leaf nodes of the regression trees of each response variable. Shown are only the most important predictors.

## 6 Discussion

### 6.1 Robustness of performance metrics

There are two important algorithmic parameters affecting drought predictions of this study. One is the threshold of drought definition, i.e. the quantile of drought index observations specifying a drought, which was set to 0.3 as commonly used in the study area among water managers. This choice affects the performance values of BSS and ROCSS. The other is the number of probability bins into which hindcasts are grouped for further analysis, affecting ROCSS but not BSS as BS was herein calculated without probability binning (see Eq. 6). Figure 9 illustrates the sensitivity of verification attributes to the two parameters. It shows that a higher drought threshold results in a more evenly running curve while a smaller threshold of 0.2 tends to be better oriented towards the reliability line and appears more variable (Fig. 9 a). This might result from the necessarily lower number of values of smaller thresholds. However, altogether general conclusions remain untouched, namely





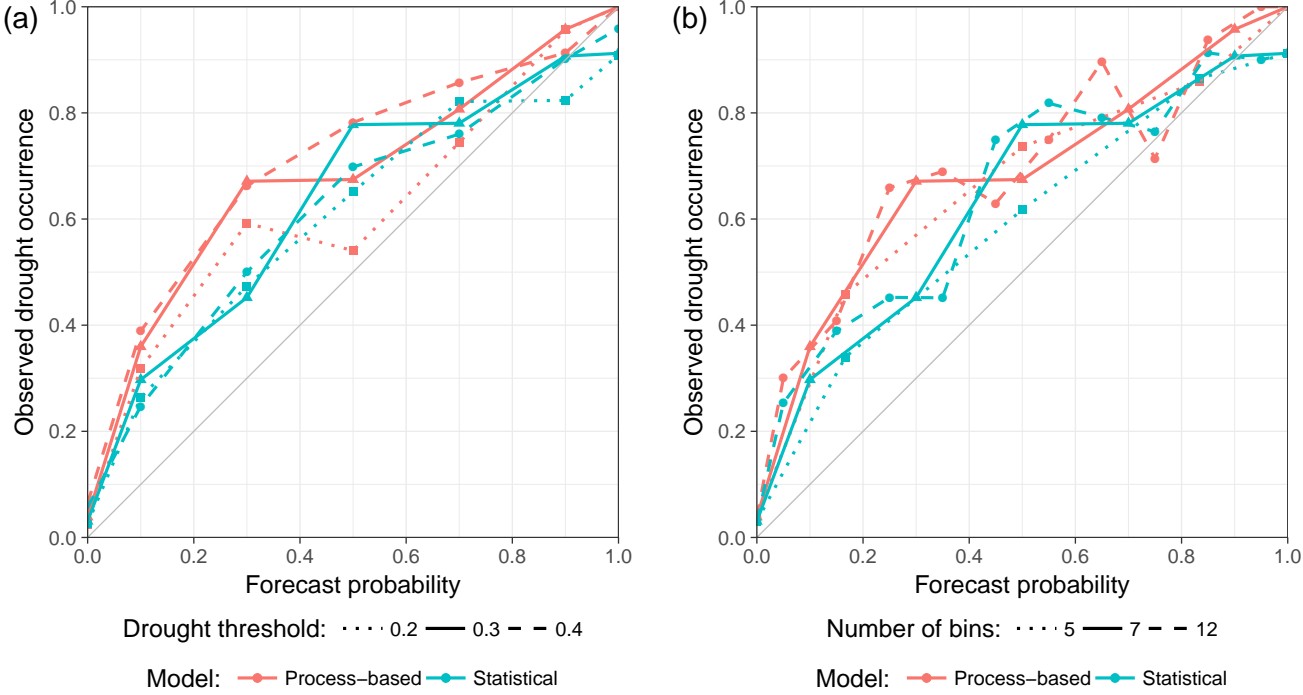

**Figure 9.** Reliability plots for different settings of (a) drought thresholds and (b) number of probability bins. Solid lines refer to the values used in this study.

underforecasting and the statistical being superior to the process-based model. Regarding the number of probability bins (Fig. 9 b), a larger value leads to a more variable curve. This effect can be attributed to the decreasing number of values per bin with increasing number of bins. For this study, it was decided to use a value of seven as it appears to be the best compromise between sufficient data availability per bin and an adequate number of bins for further calculations (namely ROCSS). Even

5 so affecting the values of ROCSS and partly BSS (not shown), it can be concluded that the somewhat arbitrary decision on a certain drought threshold and the number of bins, as long as reasonable values are chosen, does not affect the general results of the analysis.

The RMSE as accuracy measure is free of such decision parameters but is admittedly influenced in a different way. With the target variable (relative regional reservoir filling) ranging from 0 to 100 percent points, the actual maximum value tends to be

10 smaller during wet periods: the observed value (which is usually greater than zero) effectively causes the metric to be limited to about 40 to 50 percent points. This effect is reflected as the apparent performance plateau for the process-based model in Fig. 6 and is also likely to affect the results presented in Fig. 7. The effect that, when driven by hindcasts, the process-based model exhibits larger errors under dry than under wet conditions, can at least partially attributed to this issue. It seems reasonable that model simulation performance is generally better under dry conditions as can be observed when models are driven by





observations. However, as no threshold effects can be observed and the RMSE values are always considerably lower for the statistical model, this effect should not influence general conclusions of the model comparison.

## 6.2 Model comparison

In terms of simulation accuracy when driven by observations and for drought event prediction in the hindcast mode, both

models perform equally well. Hindcast accuracy, however, is substantially lower for the process-based approach. This result is well in line with findings of other studies that simple statistical model approaches often perform equally well or even better than complex process-based prediction systems, especially in tropical regions due to well exploitable correlations among meteorological and hydrological variables (Block and Rajagopalan, 2009; Hastenrath, 2012; Sittichok et al., 2018). It has to be noted, however, that the process-based approach with the WASA-SED model achieved acceptable results on monthly

(hindcasts) and even daily (calibration metrics) time scales whereas former studies in NEB reported passable results only aggregated over seasonal scales (Galvão et al., 2005; Block et al., 2009; Alves et al., 2012).

The reason for the discrepancy of model ranking between simulation and hindcast mode can be attributed to the different model structures. To illustrate this, Fig. 10 shows the average monthly changes of regional reservoir storage for the different models and modes in comparison to observations. For the simulation mode (dashed lines) it can be seen that the process-based

model, though exhibiting a constant overestimation, all in all is well in line with observations. The statistical model, however, shows a more or less constant storage change over the whole simulation horizon, resulting in over- and underestimations and, eventually, a good overall simulation performance (see Fig. 3). In hindcast mode (solid lines), for the process-based model the overestimation of storage change is much more pronounced and the peak shifted from April to March. Although the statistical model now more realistically exhibits seasonal dynamics, the general pattern still appears too smooth, which effectively results

in less deviation from observations than the output of the process-based model (Fig. 4). This indicates a strong influence of precipitation forcing on the process-based model while the statistical approach generally reacts more damped on rainfall input. Consequently, deficiencies in this forcing affect the process-based model much more. The issue of uncertainties arising from defective precipitation forcing will be later on discussed in more detail.

Despite the lower prediction performance, the process-based approach still provides benefits over the statistical model. This

includes the potential access and investigation of multiple spatially distributed hydrological variables with daily resolution, such as evapotranspiration, runoff generation, or streamflow, which were generated during the model runs. This clearly excels over the statistical model, which only yielded predictions of a single target variable. Another advantage is that model output is not only provided in a regionally and monthly aggregated manner, as for the statistical approach, but for all individual strategic reservoirs in the area as daily time series. Figure 11 illustrates that accuracies of individual reservoirs exhibit a slightly larger

variation, but the RMSE's of individual reservoirs are at a similar level as when regionally aggregated. This suggests that most of the single reservoirs can be modelled with a comparable performance as the regionally aggregated values.

A further advantage of a model such as WASA-SED is that underlying processes are directly represented. As such it can be of higher value to water managers interested not only in streamflow or reservoir level forecasts but also in the investigation of process behaviour or assessments under changing boundary conditions. Therein the model could be used in scenario analyses,





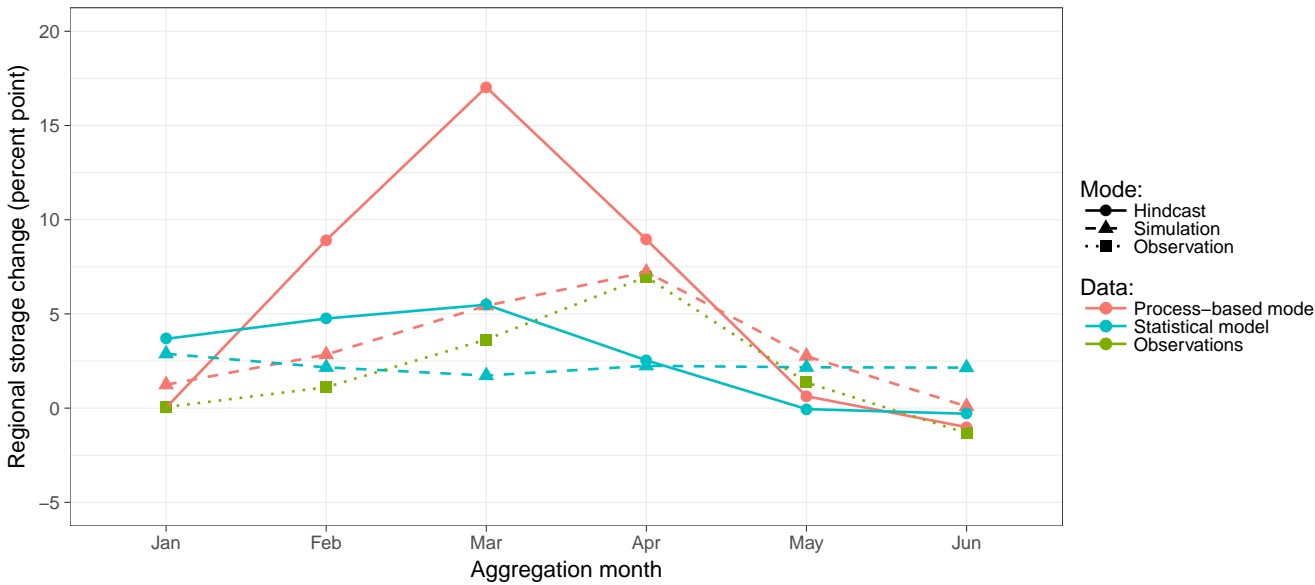

**Figure 10.** Monthly changes of regional reservoir storage averaged over all regions, years, and hindcast members for the two models and application modes in comparison to observations.

such as climate change impact assessment, or sensitivity analyses of, for instance, uncertain meteorological input to detect critical streamflow or reservoir stages. Furthermore, the model is transferable and can be easily applied in different regions and over different spatial and temporal scales, only limited by computational resources and available input data.

### 6.3 Deficiencies of the process-based simulation approach

To improve the performance of the process-based model it is first necessary to identify sources for simulation inaccuracies. It was shown that the process-based model achieved regionally different performances. A comparison of Fig. 3 and Table 2 reveals that regional bias during the calibration period is in compliance with the ranking of regional simulation errors. Moreover, although exhibiting the highest BE value, region Banabuiú is characterised by the largest bias during calibration and highest simulation and hindcast errors. As the latter is observed for both the process-based and statistical approaches, the reason

is suspected to originate from uncertainties in observations, i.e. precipitation measurements within the region, or defective reservoir level acquisition. The reason for Salgado region being out of the general pattern for the process-based model certainly originates from the different calibration procedure applied here, namely the use of streamflow measurements in contrast to reservoir dynamics as for the other regions. In addition, the region is distinct from other parts of the catchment in terms of environmental settings such as larger groundwater influence and sedimentary plateaus in the headwater area. Conversely, the

transfer of the calibrated parameters from Castanhão to the Lower Jaguaribe region seems justifiable as the simulation error was small. Overall, reservoir size largely influences both simulated storage time series and bias. Model performance, however, appears to be not superior for large reservoirs. Moreover, wetness condition in terms of antecedent rainfall sums over the last





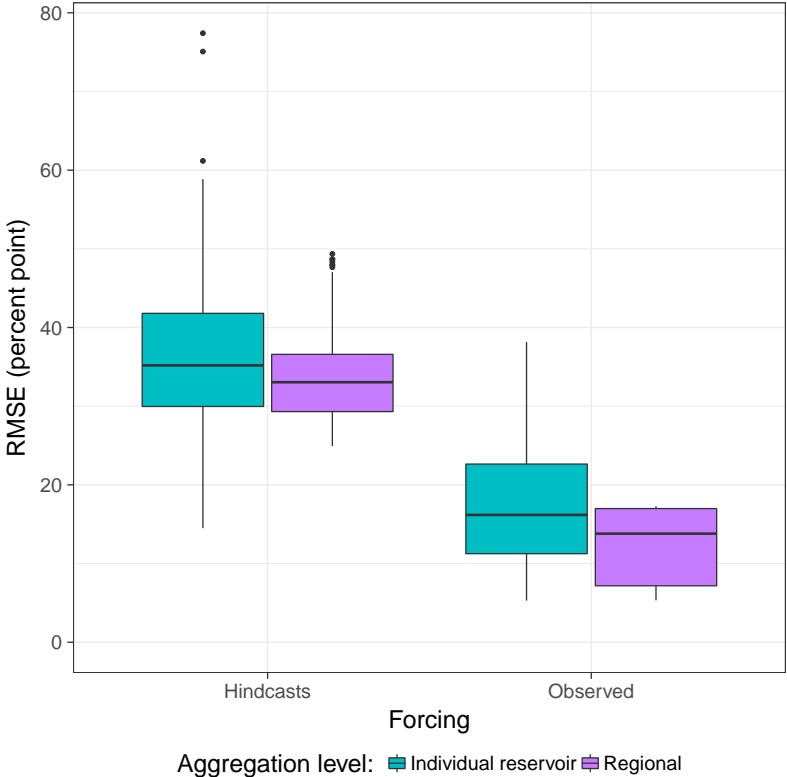

**Figure 11.** Comparison of accuracies of the process-based model for different spatial aggregation levels on a monthly time scale.

36 months is of major importance, i.e. dry conditions lead to the best model performance in terms of KGE. The latter is not surprising as rainfall in the study area is extremely heterogeneous both in space and time, usually characterised by convective heavy precipitation events with short durations. Thus, prolonged periods without rain constitute a spatially more homogenous input. Conversely, the aggregation of rainfall to daily sums and interpolation over subbasin units, on average covering an area

5   of about 700 $km^2$, must necessarily induce uncertainties. The assimilation of observed reservoir filling states at the beginning of each hindcast season is therefore a reasonable approach to improve predictions and compensate for preceding rainfall input uncertainties during the initialisation run.

### 6.4   Potential improvements

There are several options to make use of the findings of this study and improve the forecast system in upcoming applications. In

10   the presented study, observed reservoir level data were assimilated into the process-based model to correct the initial conditions for the hindcast runs by simply replacing model states by measurements. For assimilation, more formal approaches already exist such as the rich families of Kalman and particle filtering approaches (e.g. Liu and Gupta, 2007; Komma et al., 2008; Vrugt et al., 2013; Sun et al., 2016; Yan et al., 2017). These, however, require a profound quantification of both simulation and





observation uncertainties and, thus, many additional information and, moreover, significantly higher expenses in terms of data preparation, processing, and model application. Nevertheless, they hold the potential to better account for uncertainties in the observations, which were disregarded in this study, despite being considerable.

Pre-processing schemes in the context of hydrological forecasting usually focus on the improvement of rainfall predictions used as main drivers for hydrological models (e.g. Kelly and Krzysztofowicz, 2000; Reggiani and Weerts, 2008; Verkade et al., 2013). This is partly already included in the downscaling scheme applied to GCM products but may as well be further extended. The importance of rainfall forcing on model results, especially for the process-based approach, was already addressed above. A further comparison of the statistical properties (distribution of daily sums, dry/wet spell lengths) of rainfall hindcasts used in this study with observations revealed large discrepancies. Some preliminary tests suggested these to be responsible for the decreased accuracy of the process-based model hindcasts (not shown). In comparison to observations, the hindcasts contain (i) a general shift of rainfall seasonality towards the first months of the rainy season; (ii) a much lower frequency of both wet and dry periods for spell lengths up to 4 days; (iii) a lower frequency of low daily rainfall values while the number of large precipitation events is overestimated and daily extreme values are much higher; (iv) a much higher probability that a dry day follows a dry day and the probability that a wet day follows a wet day is often underestimated. These findings indicate a high potential for improvement in future applications in the study area. As a first starting point, monthly bias of hindcasts per region was corrected and both models were re-run. Figure 12 shows that this relatively simple procedure already results in a considerable decrease of RMSE for the process-based model, even though it is still higher than for the statistical approach. The improvement of drought forecast performance in terms of BSS and ROCSS is thereby less pronounced than the increase of accuracy. For the statistical model, performance metrics hardly change, which can be attributed to the smoothing effects of its model structure on regional reservoir storage identified in a previous subsection (Fig. 10).

In addition to pre-processing, post-processing approaches directly tackle the correction of streamflow forecasts by statistical means including bias correction or the estimation of an error model applied to predictions (e.g. Krzysztofowicz and Kelly, 2000; Todini, 2008; Bourdin et al., 2014; Roulin and Vannitsem, 2014). They are routinely applied in operational streamflow forecasting and, in addition to rainfall correction, could further improve model performance.

The parametrisation of the process-based model could be further improved by the use of more and different data sources. This includes, for instance, the use of satellite data to infer spatially distributed reservoir information with greater detail and more accuracy as currently available. The study area has already been of interest in ongoing research (Delgado et al., 2018) and past studies (Heine et al., 2014) addressing that issue. In addition, management plans as well as data on water abstraction and reallocation from the larger reservoir should be included in the model but were not available for the present study. Another opportunity is to increase rainfall input resolution in the model to better account for sub-daily and spatially heterogeneous precipitation. This could be done by improving the current spatial scaling of rainfall in the model to account for heterogeneous patterns and to make use of RADAR rainfall data recently made available in the area.

The combination of multiple models may provide further benefits in cases where different models show strengths in different aspects of performance (e.g. Block and Rajagopalan, 2009; Schepen and Wang, 2015). However, within this work the two employed model approaches, with respect to simulation performance achieved almost equal results and did not diverge in





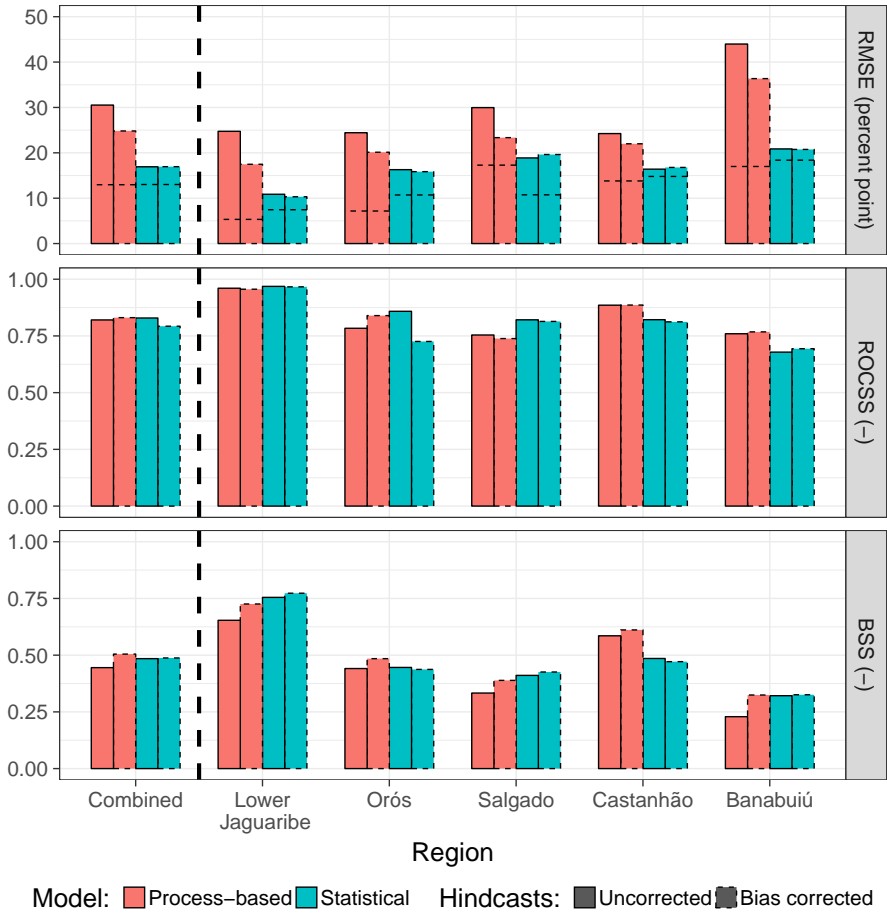

**Figure 12.** As Fig. 4 but driving hindcasts with with additional bias-correction of precipitation on the monthly scale (dotted boxes).

aspects such as lead time and antecedent moisture conditions. Thus, the combination of the two models analysed in this study is not expected to provide benefits.

## 7 Conclusions

The aim of this work was to explore options for a seasonal forecasting system of regional reservoir volume and drought occurrence with lead times up to 6 months for the semiarid northeast of Brazil. In this context, the performance of a complex process-based hydrological model was evaluated against a much simpler statistical model developed by Delgado et al. (2017) given the same meteorological forcing. The study pursued three objectives:

First, the two modelling approaches were to be investigated in terms of mere simulation performance, i.e. when driven by meteorological observations. It turned out that both models performed almost equally well. However, regional differences





exist where the process-based model achieved slightly better results in four out of five subregions. Furthermore, regional performance ranking of both models was equal in four regions. This suggests that data uncertainty of meteorological input or reservoir level observations exceeds model structural uncertainties and dictates simulation performance in the study area.

Second, the process-based model was to be verified as prediction tool in a hindcast experiment and evaluated against the statistical approach. In comparison to simulation runs with observed forcing, hindcast performance of the process-based model dropped significantly while the performance of the statistical approach decreased only to a small degree. This can be explained by the structure of the statistical approach which generally reacts more damped on precipitation inputs. Although this exhibits less realistic intra-seasonal dynamics than for the process-based model, performance metrics were eventually superior as uncertainties from precipitation hindcasts could not propagate as much to the model output. However, apart from reservoir level predictions, forecasting of mere drought occurrence works almost equally well for both approaches. The two models exhibit satisfying event resolution while slight deficiencies in terms of underforecasting were detected regarding the reliability of the hindcasts.

The third and last objective was to identify the major sources for simulation and hindcast deficiencies and provide guidelines for further improvement. In general, both models achieve better results under dry than under wet (pre-)conditions. An attempt to identify potential predictors of model performance for the process-based model revealed that reservoir size and antecedent rainfall conditions explain most of the variance of the performance metrics while variables such as current precipitation amount and daily precipitation intensity are of surprisingly low importance. However, hardly any clear patterns could be identified in which way predictors influence performance measures and, as such, no direct means could be derived, in which way the structure of the process-based model could be improved to achieve better simulation results. Also regarding the hindcasts, precipitation was identified as the most significant source of uncertainty. It was found that rainfall hindcasts from the downscaled GCM show statistical properties significantly distinct from observations. Therefore, simple approaches, such as the tested monthly regional bias correction, already result in improved hindcast accuracies. Future studies should also consider the use of more sophisticated means of pre-processing as well as post-processing approaches, such as forecast error modelling, or innovative data assimilation and data fusion approaches to correct erroneous model states.

So, what is the added value of a process-based hydrological model? When it comes to reservoir level or mere drought event prediction on regionally and monthly aggregated scales, a statistical model proved to be the better option, as computational effort is much lower and the model is easier to apply. Nevertheless, we advocate the application of a proper process-based hydrological model in case predictions on finer spatial (e.g. for individual reservoirs) and temporal scales or even more information, such as evapotranspiration or various runoff generation and concentration variables, are required. As such, only by applying a process-based hydrological model, decision makers and stakeholders can be supported to detect and understand hydrological changes in their catchments in order to make reasonable and sustainable decisions. However, further research is needed to increase the accuracy of important model drivers, i.e., in the case of dryland regions such as northeastern Brazil, first and foremost precipitation. We expect that the use of new data products, such as RADAR and satellite data along with traditional data from rainfall stations with sub-daily resolution, in combination with innovative methods of data assimilation





and data fusion provide opportunities to improve forecast accuracy of process-based hydrological models. Only in that way the time and effort of their application can be justified and allow for the exploitation of their advanced capabilities.

*Code and data availability.* Meteorological observations (except precipitation) are available from http://careyking.com/data-downloads/. Precipitation as well as raw data of meteorological hindcasts need to be requested from FUNCEME. DEM raw data can be obtained via
5  http://srtm.csi.cgiar.org/SELECTION/inputCoord.asp (tiles [horizontal/vertical]: 28/13, 28/14, 29/13, and 29/14). Reservoir data can be accessed at http://www.hidro.ce.gov.br or requested from FUNCEME. Land cover and soil maps are not publicly available.

The WASA-SED model is available at https://github.com/TillF/WASA-SED. Scripts to investigate or reproduce experiments, analyses, and compilation of plots can be accessed at https://github.com/tpilz/paper_drought_prediction_brazil.

## Appendix A:  Regression tree plots

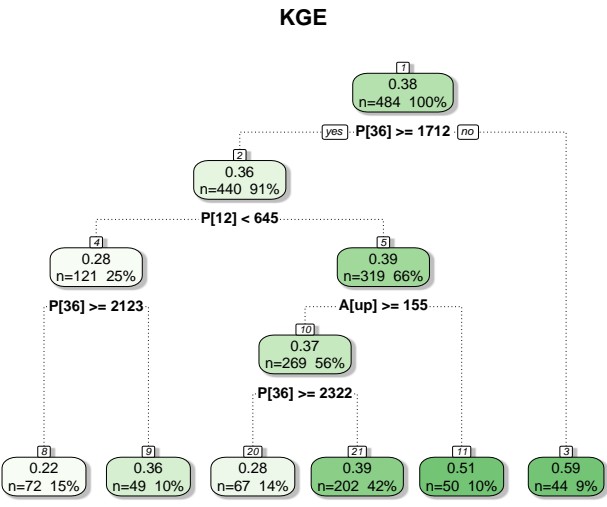

**Figure A1.** Regression tree for response variable KGE. The boxes indicate nodes, whereas the bottommost boxes are the leaf (terminal) nodes; each box shows the node number (small box on top), the fitted value of the response variable at the specific node, and the number of values reaching the node (n) including the percentage of total values; boxes are coloured according to the fitted response value, where dark green hues indicate nodes with favourable response values; below each split node, the split variable with threshold value is shown. For notation of response and predictor variables see Table 1.




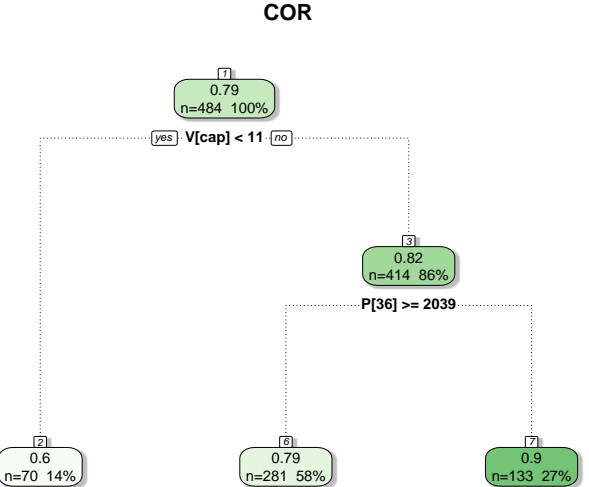

**Figure A2.** As Fig. A1 but for COR.

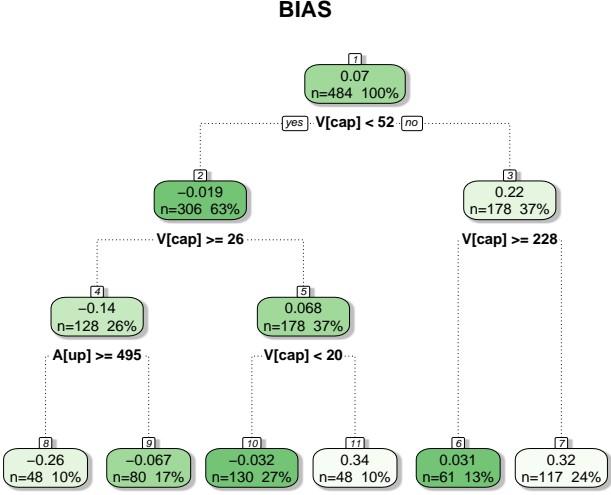

**Figure A3.** As Fig. A1 but for BIAS.

*Author contributions.* All authors contributed to the experiment design. Meteorological hindcasts were pre-processed by KV and JD. Experiments with the statistical model were conducted by SV and JD. All other experiments and analyses were conducted, figures generated, and the manuscript written by TP with support by the co-authors.

*Competing interests.* All authors declare that no competing interests are present.



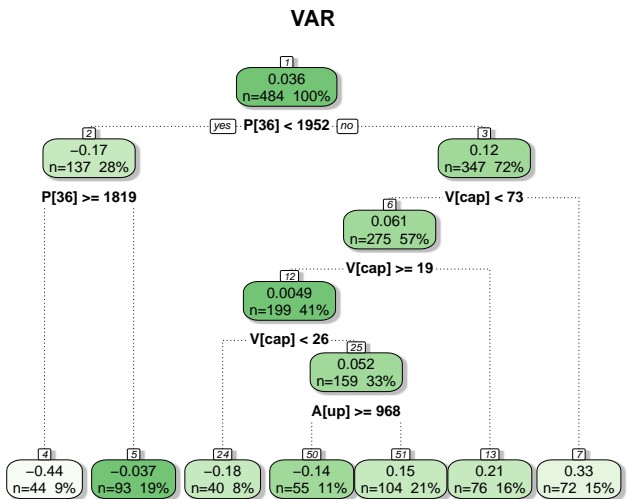

**Figure A4.** As Fig. A1 but for VAR.

*Acknowledgements.* Authors thank Gerd Bürger for his comments and discussions. TP acknowledges funding by Helmholtz graduate school GeoSim. We acknowledge the support of Deutsche Forschungsgemeinschaft (German Research Foundation) and Open Access Publication Fund of Potsdam University.



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
