# Peer review of "Seasonal drought prediction for semiarid northeast Brazil: What is the added value of a process-based hydrological model?"

_Hydrology and Earth System Sciences, 2018_

## Referee Comment (RC1) · Anonymous Referee #1 · 15 Sep 2018

Summary:

This study is a very comprehensive assessment comparing a process-based model of reservoir levels to a statistical regression approach for five river basins in northeast Brazil. The authors conducted substantial analysis to provide a thorough comparison for this region.

Major comments:

My main comment is that there is very little information regarding the statistical model used. The authors frequently reference another paper (Delgado et al, 2017), on which this work is based, however, even if the reader has not first read Delgado et al (2017),

they should be able to understand this one. Adding the equation for the regression model used, how it was designed, and how well it represented these data would greatly improve the paper. In addition, I assume that Delgado et al, 2017 was for the same location but I'm not sure that that was explicitly stated in the paper.

My second main comment is that reducing and consolidate figures to the ones that are most important and illustrative of the main points could help clarify the paper. In addition the regression tree analysis is going to be included, more information is needed. Why was this approach selected? What about the limitations of decision trees, such as the fact that they can be highly sensitive to small changes in the data or decisions about the how many nodes to allow? Were random forests, which can reduce the likelihood of overfitting, considered? Also, is it common to use regression trees for outcomes which only vary between 0 and 1 (as is done here)?

Comments on specific sections

Abstract: Line 13 Be careful about broad recommendations here. Better to be clear that these finds are specific to these particular models applied in this study and for this region. Whether a statistical approach would be better depends substantially on the specific approach used.

4.4 Hydrological modelling: At the bottom of P 8 there are a few things mentioned briefly that seem as though they could have large implications for the results. First, for lines 28-29, It seems significant to assume that reservoir losses were set to zero. Can you explain this choice a bit more? Second, for lines 29-30, using such a different metric for the Salgado region seems like it would have a large impact. It would be reassuring to have some sort of comparison to another region illustrating that this streamflow metric can accurately represent reservoir levels. For example, when I look at figure 3, I wonder if the discrepancy between Salgado and the other sites is due to this difference.

Metrics: Can you explain why you used KGE as a metric for the simulation performance

while using BE for model parameterization? Also, where does the metric in equation 2 come from? Not sure you need to add more to the paper, but it would be interesting to see a whether different quantiles (other than the 0.3 selected by decision makers) result in very different results. (Or perhaps cite any other work that might have looked at this?)

Model performance, section 5.2: It is concerning that forecast probabilities of drought are too low compared to observed occurrences. Might it be related to the issues described in Farmer, William H., and Richard M. Vogel. "On the deterministic and stochastic use of hydrologic models." Water Resources Research 52.7 (2016): 5619-5633.

Discussion: Would sections 6.1-6.3 make more sense in the results section? It seems like they include the presentation of new results rather than discussion. Adding a bit more about how the results fit within the context of previous literature would help to flesh out the discussion and conclusions.

Finally, it would be good to have some more substantial editing of the text to catch typos and clarify a confusing sentences. I would rethink the title which is potentially misleading. The "About the added value of a process-based hydrological model" might imply to some that the paper will illustrate the benefits of the process-based model. Also, being more specific that the focus is on reservoir levels rather than the broad statement of "seasonal drought" would be good.

Minor comments

P1 Line 10 "associated with" instead of "towards"?

Figure 2: I don't understand the arrows going both directions between the two skill scores boxes

P 8 Line 3 what is revision 257? Maybe you can add an appropriate citation instead of mentioning this in the text because it leaves the reader wondering a bit

P 9 line 15 "goodness of fit" measures, right?

P 9 line 30 – it would be good to define these variables in the text so the reader doesn't have to find the table.

Figure 3 might be nice to know the sample size represented by each bar

P 14 line 27 – why Jan and Feb?

Figure 5. Please explain the figure fully so that it can stand alone with the need to track down another reference.

Figure 7. for P12 (middle plot) It is interesting that RMSE decreases as conditions get more wet but increases for the statistical model. Do you have an idea of why this might be the case?

Figure 9. Explain significance of grey 1:1 line here

---

## Referee Comment (RC2) · Anonymous Referee #2 · 16 Sep 2018

This paper aims at comparing a regression tree model with a process-based hydrological model for seasonal hydrological drought prediction. While the evaluation analysis is comprehensive, critical information is missing before obtaining a conclusion that the statistical model is superior to the process-based hydrological model for predicting reservoir level. I would suggest a major revision for more solid evidences.

1. A common sense about statistical models is that they may have good performance in the training period, but degrade in the validation period. This information is missing for all results (i.e., Figures 3-12). For instance, what is the time period for the simulation in Figure 3? How about separating calibration and verification periods? This

separation is also critical for hindcast evaluation (Figure 4). In general, we have to use cross validation if there are not enough samples. And totally separating calibration and verification periods would be better. This will allow a fair comparison for statistical and dynamical models. In other words, one will never use any information in the validation period for the training or calibration. This is a basic rule for hydrological forecasting, but such important information is missing both in the text and figure captions. Both the calibration and validation results for both statistical and dynamical models should be presented. And the hindcast and forecast period should also be separated before comparison.

2. Regarding dynamical hindcast/forecast, a critical issue is whether the uncertainty in meteorological forecast affects the hydrological forecast greatly. Only one GCM (i.e., ECHAM4.6) is used in this study, which is not enough. In fact, the North American Multi-Model Ensemble (NMME) hindcast and realtime forecast data for precipitation and temperature are available for the public. There are a few global validation studies that cover the Brazil. Using multiple climate forecast models may provide an opportunity to quantify the uncertainty in the meteorological forecast. In other words, the conclusion that the statistical model outperforms the dynamical model may not be solely caused by the deficiency in hydrological model, and both the hydrological and meteorological parts should be addressed in the analysis.

3. Section "2 Terminology" could be moved to the appendix section in the end.

4. Section "4 Data and Methods" should be revised extensively. While the introduction for the models should be expanded, the introduction for the evaluation metrics could be shortened.

5. Figure 6, what does the "aggregation month" mean? Did you only carry out hindcasts from January each year? The same issue for Figure 10.

6. P2L1, "the many" -> "many"

---

## Author Comment (AC1) · 25 Sep 2018

Dear referee,

thank your very much for your feedback and constructive comments that will certainly help to improve the manuscript during the revision. As the issues raised are mainly addressing the provision of more information and clarification of unclear points, we shall deliver detailed answers within our Author's Response along with the revision of the manuscript.

However, there is one point we would like to discuss before the actual revision:

[Figure]

I would rethink the title which is potentially misleading. The "About the added value of a process-based hydrological model" might imply to some that the paper will illustrate the benefits of the process-based model. Also, being more specific that the focus is on reservoir levels rather than the broad statement of "seasonal drought" would be good.

We have considered your point and suggest the following alternative title: *Seasonal drought prediction for semiarid northeast Brazil: What is the added value of a process-based hydrological model?*

In our opinion this is a less judgmental title regarding the actual value of the process-based model and expresses the point of our manuscript. The first part of the title, however, we would rather leave as is, i.e. keeping the term "drought prediction". In our study, we explicitly defined "drought" as shortage in regional reservoir storage which, as being the actual variable of interest, has then been undergone further analyses. Consequently, we think the more general term "drought" in the title should not be misleading.

Kind regards,

Tobias Pilz (on behalf of all co-authors)

---

## Author Comment (AC2) · 25 Sep 2018

Dear referee,

thank you for your detailed and fruitful comments on our manuscript. All your critics shall be fully addressed in our Author's Response along with the revised manuscript. In this response we merely want to further discuss two specific points raised in your review.

1. A common sense about statistical models is that they may have good performance in the training period, but degrade in the validation period. This

information is missing for all results (i.e., Figures 3-12). For instance, what is the time period for the simulation in Figure 3? How about separating calibration and verification periods? This separation is also critical for hindcast evaluation (Figure 4). In general, we have to use cross validation if there are not enough samples. And totally separating calibration and verification periods would be better. This will allow a fair comparison for statistical and dynamical models. In other words, one will never use any information in the validation period for the training or calibration. This is a basic rule for hydrological forecasting, but such important information is missing both in the text and figure captions. Both the calibration and validation results for both statistical and dynamical models should be presented. And the hindcast and forecast period should also be separated before comparison.

We are aware of the common procedure of calibration and validation using independent time periods in typical applications of hydrological simulation. However, we do not see a conflict to the procedure in our study, which included:

- **calibration** of parameters within time period 2003 to 2010 **forced by observations**

- model state initialisation 1980 to 2014 forced by observations

- assimilation of observed reservoir levels at the beginning of each year (i.e. **correction of model state**)

- **hindcasts** 1981 to 2014 (Jan to Jun) **forced by GCM predictions**

Hence, calibration and hindcast period, even though the time periods overlap, differ in a) the forcing, and b) model's reservoir storage which was updated by measurements.

[Figure]

Consequently, we think the hindcast and calibration runs can be regarded as independent. However, we admit the lengthy Methods section and the many Figures to be confusing at some points and shall seek for more clarification on that issue during the manuscript revision.

The other point to discuss is:

> 2. Regarding dynamical hindcast/forecast, a critical issue is whether the uncertainty in meteorological forecast affects the hydrological forecast greatly. Only one GCM (i.e., ECHAM4.6) is used in this study, which is not enough. In fact, the North American Multi-Model Ensemble (NMME) hindcast and realtime forecast data for precipitation and temperature are available for the public. There are a few global validation studies that cover the Brazil. Using multiple climate forecast models may provide an opportunity to quantify the uncertainty in the meteorological forecast. In other words, the conclusion that the statistical model outperforms the dynamical model may not be solely caused by the deficiency in hydrological model, and both the hydrological and meteorological parts should be addressed in the analysis.

We absolutely agree that precipitation forcing provided by the GCM certainly is a great source of uncertainty and interesting for further analyses. However, we discussed the use of further GCMs already during the conception of our study but finally decided against it because:

- we wanted to focus on the comparison of using two different types of hydrological models for drought forecasting

- we considered an ensemble of 20 realisations of ECHAM4.6, reflecting uncertainty arising from a specific GCM

- forcing uncertainty was explicitly addressed and discussed and in the paper, albeit in a simplified manner, see subsections 6.2 (p. 21 lines 12 to 23) and 6.4 (p. 24 lines 4 to 20)

- the paper is already rather long and extending it by a further aspect would make it even more complicated to understand

Therefore, we think properly addressing forcing uncertainty for drought forecasting would rather fill another full paper and should be investigated more comprehensively in an independent study. We suggest not to extend the current scope of the paper.

Kind regards,

Tobias Pilz (on behalf of all co-authors)

---

## Author Response (AR1)

**Author's response**

First of all, we would like to thank the two anonymous reviewers for their helpful suggestions and comments. During the revision of the manuscript we tried to address the issues raised by the reviewers as much as possible. A marked-up version of the revised manuscript is appended at the end of this document. In the following we list the Referee comments (indented and in italics) together with our responses. Note that, unless otherwise stated, all section and figure references refer to the revised manuscript and might deviate from the old version. We believe the quality of the manuscript has now been further improved and hope to have satisfactorily addressed all raised concerns.

**RC1 by Anonymous Referee #1**

> *My main comment is that there is very little information regarding the statistical model used. The authors frequently reference another paper (Delgado et al, 2017), on which this work is based, however, even if the reader has not first read Delgado et al (2017), they should be able to understand this one. Adding the equation for the regression model used, how it was designed, and how well it represented these data would greatly improve the paper. In addition, I assume that Delgado et al, 2017 was for the same location but I'm not sure that that was explicitly stated in the paper.*

Section 3.5 was extended and now provides more detailed information on the statistical model. Besides, Table 2 was added showing the regionally derived equations and goodness of fit measures.

> *In addition the regression tree analysis is going to be included, more information is needed. Why was this approach selected? What about the limitations of decision trees, such as the fact that they can be highly sensitive to small changes in the data or decisions about the how many nodes to allow? Were random forests, which can reduce the likelihood of overfitting, considered?*

We acknowledge the suggestion of the reviewer to consider random forests instead of traditional regression tree analysis. Moreover, we found an improved implementation which can better handle correlation among predictors (which is the case in our example). Therefore, we replaced the regression tree approach by random forest and adapted the methods and results sections accordingly. Thereby, Table 3 (variable importances) of the old version has been replaced by a barplot (Fig. 8 in the revised manuscript) and Fig. 9 was adapted. Note, however, that results did not change significantly.

> *Also, is it common to use regression trees for outcomes which only vary between 0 and 1 (as is done here)?*

Regression trees are used for continuous numeric response variables. As far as we can judge, a bounded response variable (e.g. correlation $\in [-1, 1]$ or KGE $\in (-\infty, 1]$) can be used without a problem, as regression trees generally cannot be used for extrapolation (i.e. the result of any prediction will always be within the range of the training dataset of the response variable).

> *Abstract: Line 13 Be careful about broad recommendations here. Better to be clear that these finds are specific to these particular models applied in this study and for this region. Whether a statistical approach would be better depends substantially on the specific approach used.*

The general recommendation was replaced by a more specific formulation, which keeps the present case study in mind.

> *4.4 Hydrological modelling: At the bottom of P 8 there are a few things mentioned briefly that seem as though they could have large implications for the results. First, for lines 28-29, It seems significant to assume that reservoir losses were set to zero. Can you explain this choice a bit more?*

Admittedly, the description here was incomplete and not easy to understand. The idea was to only consider increases of reservoir volume during calibration which are caused by runoff draining into the reservoir. Volume losses, on the other hand, are largely determined by water withdrawals, overspill and artificial release, for which no (withdrawals, overspill) or incomplete (release) data were available. Therefore, to exclude such uncertain management influence from calibration, negative volumes changes were set to zero (in the observations as well as simulations) to effectively exclude these effects from calibration.

*Second, for lines 29-30, using such a different metric for the Salgado region seems like it would have a large impact. It would be reassuring to have some sort of comparison to another region illustrating that this streamflow metric can accurately represent reservoir levels. For example, when I look at figure 3, I wonder if the discrepancy between Salgado and the other sites is due to this difference.*

For clarification: for Salgado region river discharge was used as target variable for calibration, while for the other regions daily increases of reservoir volumes were used. The metric for the quantification of model performance, however, was the same for all regions (BE defined by eq. 1).

It is for sure likely that the different target variable for calibration of the Salgado region has some influence on the calibration and eventually on the forecast results. However, the Salgado region is also characterised by different hydrological conditions (more groundwater influence due to different geology) which might as well be the reason for the discrepancy in Fig. 3. This issue is discussed in the discussion Sect. 6.3 (p. 22, lines 11 ff. first manuscript version).

We hope this clarification along with the adaptations to Sect. 3.4.2 meet the reviewer's concerns.

*Metrics: Can you explain why you used KGE as a metric for the simulation performance while using BE for model parameterization?*

BE was used as performance measure for calibration as it is easy to interpret because of its relation to the common Nash–Sutcliffe metric, but it is better suited for catchments with strong seasonal dynamics. For the detailed analysis of simulation performance we decided not to focus on just a single metric, as was sufficient for the automated calibration algorithm, but on different performance measures which evaluate different aspects. The KGE appeared to be most suitable to us, as it considers different aspects, namely correlation, bias, and standard deviation, which were then analysed both individually and in combination (the KGE value).

We adapted the manuscript to clarify our intention and justify the use of different metrics (Sect. 3.4.3).

*Also, where does the metric in equation 2 come from? Not sure you need to add more to the paper, but it would be interesting to see a whether different quantiles (other than the 0.3 selected by decision makers) result in very different results. (Or perhaps cite any other work that might have looked at this?)*

Water resources in the study area are to a large degree formed by many small and some large strategic surface reservoirs. Droughts in this area are therefore commonly characterised by a shortage of these water resources. Consequently we defined the monthly regional water storage as indicator (eq. 2) and a threshold value to identify droughts (eq. 3) as straightforward and intuitive procedure. There is no explicit literature reference, the procedure was rather chosen to meet the local conditions as best as possible (and it was also used in the paper of Delgado et al. (2018b), on which our study is partly based). The impact of different quantiles for drought definition is discussed in Sect. 5.1 (and shown in Fig. 10) of the paper.

Section 3.6.1 was rephrased to clarify the issue and a reference to the discussion section was added regarding the quantile decision. With this we hope to have adequately addressed the reviewer's concerns.

*Model performance, section 5.2: It is concerning that forecast probabilities of drought are too low compared to observed occurrences. Might it be related to the issues described in Farmer, William H., and Richard M. Vogel. "On the deterministic and stochastic use of hydrologic models." Water Resources Research 52.7 (2016): 5619-5633.*

The paper by Farmer and Vogel (2016) illustrates the importance of explicitly considering the characteristics of residuals from model simulations as otherwise, while the overall model performance might by acceptable, there can be a large bias in simulated streamflow statistics, e.g. when focussing on extremes such as floods or droughts. We briefly discussed that issue under the term *post-processing* methods in the discussions (Sect. 6.4, p. 24, l. 21 ff. first manuscript version). For our study, we expect deficiencies in rainfall predictions to be the main reason for low prediction performances of reservoir levels and drought events. However, it would be interesting to investigate and compare different error sources and correction schemes in future studies. We therefore thank the reviewer for the remark and extended Sect. 5.4 accordingly.

*Discussion: Would sections 6.1-6.3 make more sense in the results section? It seems like they include the presentation of new results rather than discussion. Adding a bit more about how the results fit within the context of previous literature would help to flesh out the discussion and conclusions.*

We admit that the discussion section contains four figures, which might appear a bit unconventional. However, all figures were added to support our argumentation in the discussion. We believe this makes our points better understandable and would therefore keep the figures. Another option would be to merge the discussion and results sections. However, as some parts of the discussion go well beyond what is presented in the results, in our opinion this would leave the paper less understandable. Regarding the context of previous literature, there are currently 23 citations within the discussion section, whereas six are related to similar studies as presented here (Sect. 5.2) and the rest referring to possibilities of future improvement of our approach (Sect. 5.4). Apart from the comparison to previous literature, we admittedly focus on the discussion of our results in way to explain deficiencies in our results and how these can be overcome in future applications. A reason for this also is that (apart from the already included citations) we are not aware of further studies which are comparable to our approach. Therefore we would ask to leave the discussion sections as is (with some minor adjustments to improve readability).

*I would rethink the title which is potentially misleading. The "About the added value of a process-based hydrological model" might imply to some that the paper will illustrate the benefits of the process-based model. Also, being more specific that the focus is on reservoir levels rather than the broad statement of "seasonal drought" would be good.*

As already stated in our reply during the discussion process ('Reply to Referee #1: Suggestion for an alternative title' from 25 Sep 2018) we changed the title of the manuscript to *Seasonal drought prediction for semiarid northeast Brazil: What is the added value of a process-based hydrological model?*. The first part of the title we decided to leave as is, i.e. keeping the term "drought prediction". The reason is that, for our study, we explicitly defined the term "drought" as shortage in regional reservoir storage which, as being the actual variable of interest, has then been undergone further analyses. Therefore, in our opinion the general term "drought" in the title should not be misleading.

*P 14 line 27 – why Jan and Feb?*

*Figure 7. for P12 (middle plot) It is interesting that RMSE decreases as conditions get more wet but increases for the statistical model. Do you have an idea of why this might be the case?*

This is probably caused by the fact that RMSE of regional reservoir filling in practice is larger during dry than during wet periods. Besides, for the process-based model, RMSE tends to vary more among the 20 ensemble members because the simulated dynamics (especially during filling up of empty reservoirs) is more diverse in comparison to the rather smooth and inert dynamics of the statistical approach, which, though the dynamics appearing too smooth, exhibits a lower RMSE. Therefore, for the statistical model the differences between hindcast and simulation mode are also much less pronounced than for the process-based model. This is discussed in more detail in the discussion section (p. 20 l. 8 ff. and p. 21 l. 12 ff. first manuscript version). To avoid confusions, the discussion section has been adapted and cross references were added to clarify the links betweens figures in the results and explanations in the discussion sections (see Sects. 5.1, 2. paragraph and 5.2, 2. paragraph).

*P1 Line 10 "associated with" instead of "towards"?*

*Figure 2: I don't understand the arrows going both directions between the two skill scores boxes*

*P 8 Line 3 what is revision 257? Maybe you can add an appropriate citation instead of mentioning this in the text because it leaves the reader wondering a bit*

*P 9 line 15 "goodness of fit" measures, right?*

*P 9 line 30 – it would be good to define these variables in the text so the reader doesn't have to find the table.*

*Figure 3 might be nice to know the sample size represented by each bar*

*Figure 5. Please explain the figure fully so that it can stand alone with the need to track down another reference.*

*Figure 9. Explain significance of grey 1:1 line here*

The mentioned remarks were included and the respective lines or figures adapted accordingly.

**RC2 by Anonymous Referee #2**

*1. A common sense about statistical models is that they may have good performance in the training period, but degrade in the validation period. This information is missing for all results (i.e., Figures 3-12). For instance, what is the time period for the simulation in Figure 3? How about separating calibration and verification periods? This separation is also critical for hindcast evaluation (Figure 4). In general, we have to use cross validation if there are not enough samples. And totally separating calibration and verification periods would be better. This will allow a fair comparison for statistical and dynamical models. In other words, one will never use any information in the validation period for the training or calibration. This is a basic rule for hydrological forecasting, but such important information is missing both in the text and figure captions. Both the calibration and validation results for both statistical and dynamical models should be presented. And the hindcast and forecast period should also be separated before comparison.*

This remark has been addressed during manuscript discussion (see 'Reply to Referee #2: clarification on calibration period and discussion of further GCM employment' from 25 Sep 2018). As this remained unanswered we want to bring forward our arguments again:

We are aware of the common procedure of calibration and validation using independent time periods in typical applications of hydrological simulation. However, we do not see a conflict to the procedure in our study, which included:

– **calibration** of parameters within time period 2003 to 2010 **forced by observations**

– model state initialisation 1980 to 2014 forced by observations

– assimilation of observed reservoir levels at the beginning of each year (i.e. **correction of model state**)

– **hindcasts** 1981 to 2014 (Jan to Jun) **forced by GCM predictions**

Hence, calibration and hindcast period, even though the time periods overlap, differ in a) the forcing, and b) model's reservoir storage which was updated by measurements. Consequently, we think the hindcast and calibration runs can be regarded as independent.

With the changes to the manuscript we hope to have clarified ambiguities (e.g. regarding Fig. 3).

*2. Regarding dynamical hindcast/forecast, a critical issue is whether the uncertainty in meteorological forecast affects the hydrological forecast greatly. Only one GCM (i.e., ECHAM4.6) is used in this study, which is not enough. In fact, the North American Multi-Model Ensemble (NMME) hindcast and realtime forecast data for precipitation and temperature are available for the public. There are a few global validation studies that cover the Brazil. Using multiple climate forecast models may provide an opportunity to quantify the uncertainty in the meteorological forecast. In other words, the conclusion that the statistical model outperforms the dynamical model may not be solely caused by the deficiency in hydrological model, and both the hydrological and meteorological parts should be addressed in the analysis.*

Again, this was addressed during manuscript discussion:

We absolutely agree that precipitation forcing provided by the GCM certainly is a great source of uncertainty and interesting for further analyses. However, we discussed the use of further GCMs already during the conception of our study but finally decided against it because:

– we wanted to focus on the comparison of using two different types of hydrological models for drought forecasting

– we considered an ensemble of 20 realisations of ECHAM4.6, reflecting uncertainty arising from a specific GCM

– forcing uncertainty is explicitly addressed and discussed and in the paper, albeit in a simplified manner, see subsections 5.2 (2. paragraph) and 5.4 (2. paragraph)

– the paper is already rather long and extending it by a further aspect would make it even more complicated to understand

Therefore, we think properly addressing forcing uncertainty for drought forecasting would rather fill another full paper and should be investigated more comprehensively in an independent study. We suggest not to extend the current scope of the paper.

*3. Section "2 Terminology" could be moved to the appendix section in the end.*

*4. Section "4 Data and Methods" should be revised extensively. While the introduction for the models should be expanded, the introduction for the evaluation metrics could be shortened.*

*6. P2L1, "the many" -> "many"*

We acknowledge the remarks and adapted the respective sections accordingly. We belief this improved the readability of the manuscript and made text and figures better understandable.

*5. Figure 6, what does the "aggregation month" mean? Did you only carry out hindcasts from January each year? The same issue for Figure 10.*

We carried out hindcasts separately for each rainy season (January to June) over the hindcast period (1981 to 2014). In Figs. 6 and 10 results were aggregated for each month of the rainy season over the full period. We hope in the revised manuscript our approach became more obvious.

[revised manuscript text omitted]

---

## Author Response (AR2)

**Author's response**

We thank the two anonymous reviewers for the positive feedback on our revised manuscript. The further minor suggestions were considered and the manuscript revised accordingly. We believe this further enhanced the general understanding of the paper. In the following, the raised issues will be answered, followed by the improved version of our manuscript with highlighted changes (relative to the revised manuscript).

> *P 9 line 25 "an regression approach", an should be "a"*
>
> *P 20 line 4 check for missing word*

The lines were corrected.

> *P 9 line 28. This description is a little confusing. Is it really the "effect of each predictor" or the correlation observed in the random forest?*
>
> *P 16 How are the predictors related to the performance metrics within the random forest approach?*
>
> *Figure 8. Would be good to expand the caption here and explain how these "Predictor importances for each response variable" were determined and a definition for what they are important for.*

We assume the reviewer meant P8 line 20 instead of P 9 line 28 as the latter would better fit into the context of the question (we believe the question was addressing the random forest approach instead of the statistical model). Sections 3.4.3 and 4.3.2, and the caption of Figure 8 were slightly re-phrased and extended in order to better describe "predictor importance" as it was used in this study.

> *P 9 line 26 "They fitted" should be "they fit" but did you also do this? If so, focus on your application of the approach to make this clear.*
>
> *P 9, line 27. The sentence starting with "As response variable…" is a little confusing. Maybe a comma after variable and remove "for model fit". Also, they are other possible predictors, so clarify that these are possible predictors that were examined in this study.*
>
> *P 9, line 30 – Would help to provide a reference for this a heuristic search algorithm or name the particular program that was used. Does the algorithm determine "best" based on a particular goodness of fit metric?*
>
> *Table 2. Adjusted R2 is a more fair metric to report because it accounts for the number of predictor variables in the regression equation. I realize that this table comes from the Delgado et al paper, but I would switch to adjusted R2 if possible. This is especially important given the large number of predictor variables in these equations. For example, see the discussion here: http://thestatsgeek.com/2013/10/28/r-squared-and-adjusted-r-squared/ Was any cross validation was done with these equations in Delgado et al? To ensure that these equations aren't overfitting the data, this would be very important.*

Section 3.5 has been rephrased addressing the raised issues. Regarding the use of R2 vs. "adjusted R2" it should be noted that the number of observations used for model fit was rather large (monthly values for the period 1986 to 2014 less some missing values, i.e. up to 348 values) in comparison to the number of predictors (max. six). Therefore, the difference between R2 and Adjusted R2 is negligible (I roughly estimated the difference to be less than 0.05 for the R2 values in Table 2).

[revised manuscript text omitted]

---

## Author Response (AR3)

**Author's response**

**Dear Dr. Shukla,**

we are sorry to confess that, while focussing on the reviewer comment, your concerns regarding the broader implications of the analysis inadvertently remained unattended. To address this issue, we extended the discussion (i.e. we added the new sub-section 5.5) and conclusions. We hope your concerns could be eliminated. What follows is the revised manuscript with highlighted changes.

**Seasonal drought prediction for semiarid northeast Brazil: What is the added value of a process-based hydrological model?**

Tobias Pilz1, José Miguel Delgado1, Sebastian Voss1, Klaus Vormoor1, Till Francke1, Alexandre Cunha Costa2, Eduardo Martins3, and Axel Bronstert1

[revised manuscript text omitted]